DOI: 10.1038/s41467-018-07324-5　　**OPEN**

# A CRISPR–Cas9-triggered strand displacement amplification method for ultrasensitive DNA detection

Wenhua Zhou [1], Li Hu[1], Liming Ying [2], Zhen Zhao[1], Paul K. Chu [3,4] & Xue-Feng Yu[1]

Although polymerase chain reaction (PCR) is the most widely used method for DNA amplification, the requirement of thermocycling limits its non-laboratory applications. Isothermal DNA amplification techniques are hence valuable for on-site diagnostic applications in place of traditional PCR. Here we describe a true isothermal approach for amplifying and detecting double-stranded DNA based on a CRISPR–Cas9-triggered nicking endonuclease-mediated Strand Displacement Amplification method (namely CRISDA). CRISDA takes advantage of the high sensitivity/specificity and unique conformational rearrangements of CRISPR effectors in recognizing the target DNA. In combination with a peptide nucleic acid (PNA) invasion-mediated endpoint measurement, the method exhibits attomolar sensitivity and single-nucleotide specificity in detection of various DNA targets under a complex sample background. Additionally, by integrating the technique with a Cas9-mediated target enrichment approach, CRISDA exhibits sub-attomolar sensitivity. In summary, CRISDA is a powerful isothermal tool for ultrasensitive and specific detection of nucleic acids in point-of-care diagnostics and field analyses.

[1] Center for Biomedical Materials and Interfaces, Shenzhen Institutes of Advanced Technology, Chinese Academy of Sciences, Shenzhen 518055, China.
[2] Faculty of Medicine, Molecular Medicine, National Heart and Lung Institute, Imperial College London, London SW7 2AZ, UK. [3] Department of Physics, City University of Hong Kong, Hong Kong, China. [4] Department of Materials Science and Engineering, City University of Hong Kong, Hong Kong, China. These authors contributed equally: Wenhua Zhou, Li Hu. Correspondence and requests for materials should be addressed to X.-F.Y. (email: xf.yu@siat.ac.cn)

O wing to the limitation of power-hungry thermocycling in conventional polymerase chain reactions (PCRs), efficient isothermal DNA amplification tools are of great importance to basic research as well as diagnostic/monitoring applications. Although several methods such as the padlock/molecular inversion probes (MIP)-mediated methods, nucleic acid sequence-based amplification (NASBA), strand displacement amplification (SDA), loop-mediated isothermal amplification (LAMP), helicase-dependent amplification (HDA), and recombinase polymerase amplification (RPA) have been proposed[1–7], they suffer from trade-offs with regard to sensitivity, specificity, simplicity, and cost. Moreover, some of these techniques still require an initial heat-denaturation step to unwind double-stranded DNA (dsDNA) targets thereby further limiting their applications. In this respect, a true isothermal DNA amplification/detection method with attomolar sensitivity, single-base specificity, and simple reaction scheme is highly desirable.

The CRISPR–Cas system (Clustered Regularly Interspaced Short Palindromic Repeat and CRISPR-Associated Protein) originally derived from adaptive immune systems in bacteria and archaea against invading nucleic acid components[8], has become a prominent tool in transcription regulation, genome editing, and in situ DNA/RNA detection[9–14]. In particular, on account of the simplicity and high sensitivity/specificity, various CRISPR effectors including Cas9, Cas12a, and Cas13a/b have been explored and the prospect of CRISPR-based nucleic acids diagnostics (CRISPR-Dx) is encouraging[15–17]. However, all the CRISRP-Dx approaches reported so far require an initial amplification step such as PCR, NASBA, and RPA to specifically amplify target nucleic acids and CRISPR effectors are only used in endpoint analyses. In other words, the performance of these CRISRP-Dx approaches is mainly determined by the initial amplification, while the superior sensitivity and stringent PAM recognition of CRISPR systems have not been fully exploited.

Similar to most monoclonal antibodies, the equilibrium dissociation constants of CRISPR effectors towards targets are in the nanomolar regime ($10^{-8}$–$10^{-10}$ M)[18–21], suggesting that they are highly favorable for sensitive recognition of nucleic acids. The low tolerance in the PAM-proximal mismatches[22–24] also indicates that CRISPR effectors may provide sufficient specificity for the detection of single-nucleotide polymorphisms (SNPs). Moreover, previous investigation has revealed that the non-target strand of DNA targets in CRISPR ribonucleoprotein complexes are fully unwound and exposed to the environment (formation of R-loop), which are susceptible to hydrolysis[25,26]. This unique conformational rearrangement may provide an ideal targeting site for various isothermal amplification techniques with enhanced robustness, specificity, and sensitivity due to the intrinsic properties of CRISPR effectors. For example, when combined with the exponential amplification reaction (EXPAR) and rolling cycle amplification (RCA), the CRISPR effector, Cas9, has been successfully applied in the efficient pre-screening of sgRNAs[27] and sensitive in situ genomic loci detection[28], respectively. Thus, this CRISPR effectors-triggered strategy has great potentials to be applied in different situations.

Herein, we report a CRISPR–Cas9-triggered nicking endonuclease-mediated SDA method (abbreviated as CRISDA) for sensitive amplification and detection of double-stranded DNA (dsDNA) and this technique takes full advantage of the high sensitivity, stringent specificity, and unique conformational rearrangements of CRISPR effectors in recognizing the target DNA. In combination with a robust peptide nucleic acid (PNA) invasion-mediated endpoint measurement[29,30], the method achieves attomolar sensitivity and single-base specificity in the presence of a complex sample background. After conducting a series of proof-of-concept investigations, the attomolar sensitivity of CRISDA is examined by specific amplification and detection towards target DNA fragments as well as target regions in the genomes of human and genetically modified soybean MON87705. The single-nucleotide specificity of CRISDA is evaluated by breast cancer-associated SNPs genotyping among various cell lines and finally, the versatility of CRISDA is demonstrated by integrating the technique with a Cas9-mediated target enrichment approach exhibiting sub-attomolar sensitivity. CRISDA is demonstrated to be a powerful tool in ultrasensitive detection and diagnosis of nucleic acids.

## Results

**Rational design of CRISDA.** The schematic reaction mechanism of CRISDA is illustrated in Fig. 1. The well-characterized *S. pyogenes* Cas9 with a mutation of HNH catalytic residue (spy-Cas9H840A nickase, named as Cas9) is used as a model. First, a pair of sgRNAs ($sg_{UPS/DNS}$) is designed to program the Cas9 ribonucleoprotein recognizing each border of the target DNA and inducing a pair of nicks in both non-target strands (Step 1). Subsequently, a pair of primers is introduced to initiate the SDA reactions (Initiating Primer pair, $IP_{UPS/DNS}$) and hybridized to the exposed non-target strands (Step 2). Each IP primer contains a 5′ Nb.BbvCI endonuclease nicking site, a middle hybridization region complementary to the exposed non-target strand, and a 3′ overhang complementary to the double-stranded region of the non-target strand. After adding the SDA mixtures, DNA polymerases (Klenow Fragment 3′ → 5′ exo⁻, KF) extend IP primers along the non-target strands with the help of single-stranded DNA (ssDNA) binding protein TP32 (SSB). Meanwhile, the exposed non-target strands are also extended by DNA polymerases from the nicking site induced by Cas9 to the 5′ end of annealed IP primers, generating a double-stranded Nb.BbvCI endonuclease nicking site. Additionally, Nb.BbvCI nickase, KF exo⁻ polymerase, and SSB work together to give linear strand displacement reactions along the target DNA to the opposite Cas9 binding site (Step 3). Afterwards, the linearly replaced single strands, Strand-For and Strand-Rev, are annealed to the primers, $IP_{DNS}$ and $IP_{UPS}$, respectively, which further induce exponential SDA of the selected target sequence within 90 min giving products 1–3 (Step 4). Subsequent PAGE analysis reveals two bands at the position of amplicon, where one corresponds to products 1 and 2 with the same length and the other one is product 3, which is about 20 bp shorter, as has been observed previously[31]. The whole process is carried out at a constant temperature between 25 and 40 °C. Finally, the biotin-labeled PNA and Cy5-labeled PNA probes targeting the middle region of the amplicon are added and the amplified target DNA is quantitatively determined by the fluorescence intensity of Cy5 after a simple magnetic pull-down (Step 5).

**CRISPR–Cas9-triggered linear SDA reactions.** To prove the principle, we first investigate whether the designed IP primer can effectively bind to the exposed region of non-target strand in the DNA–Cas9 ribonucleoprotein complex and provide an initiating site for linear SDA reactions as the first step in CRISDA reactions. After incubating a Cy5-labeled 269 bp DNA fragment from pGL3 plasmid (Fragment of Target 1 from plasmid, namely pTF1-Cy5, 100 nM) with Cas9–sgRNA ribonucleoprotein (Cas9 and $sg_{pTF1-DNS}$, both at 50 nM) targeting a downstream site (DNS site) for 10 min at 37 °C, a 5′ Cy3-labeled IP primer ($IP_{T1-DNS}$-Cy3, 100 nM) hybridized to the exposed region of the non-target strand is added and incubated for 5 min. Fluorescent electrophoresis mobility shift assays (EMSA) reveal a colocalized Cy5 and Cy3 species (Species II) with slower mobility than Cy5–DNA–Cas9 complex (Species I) indicating successful

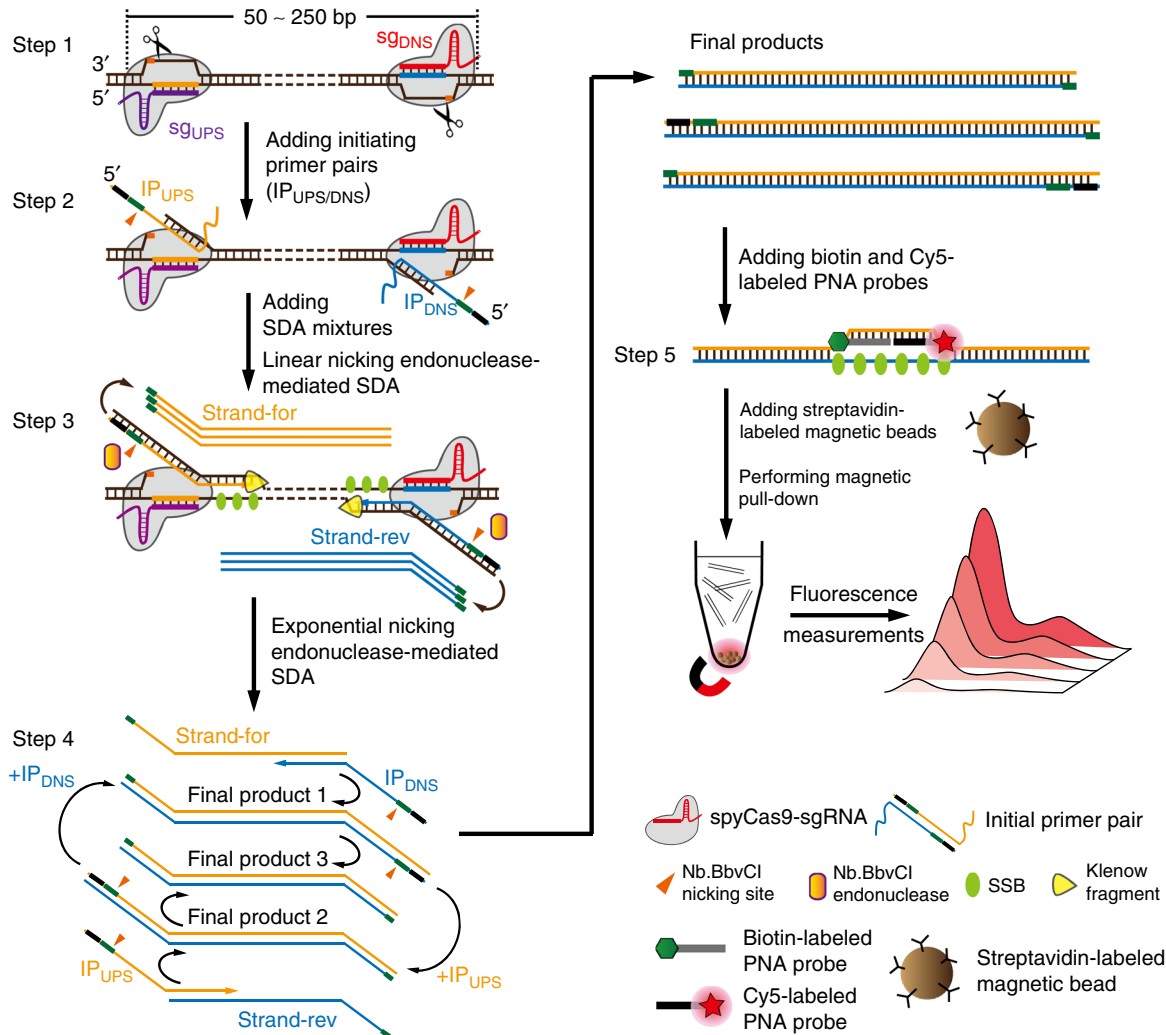

**Fig. 1** Schematic reaction mechanism of CRISDA. Step 1: A pair of Cas9 ribonucleoproteins is programmed to recognize each border of the target DNA and to induce a pair of nicks in both non-target strands. Step 2: A pair of IP primers is introduced and hybridized to the exposed non-target strands. Step 3: After adding SDA mixtures containing KF polymerase (3′– > 5′ exo⁻), Nb.BbvCI nikase, and single-stranded DNA binding protein TP32 (SSB), linear SDA is initiated from the binding sites of IP primers, giving linearly replaced single strands, the Strand-for and Strand-rev. Step 4: The products, Strand-For and Strand-Rev, are annealed again to the IP primers, which further induce exponential SDA of the selected target sequence. Step 5: The amplicons are quantitatively determined by a PNA invasion-mediated endpoint measurement via magnetic pull-down and fluorescence measurements. The well-characterized *S. pyogenes* Cas9 with a mutation of HNH catalytic residue (spyCas9H840A nickase) is used as a model. *Two bands will be observed in PAGE analyses, where one corresponds to the final products 1 and 2 with the same length and the other one is product 3

formation of IP–DNA–Cas9 complexes (Fig. 2a, b). To explore initiation of SDA from the IP primer, the SDA mixtures containing KF polymerase (0.8 U μL⁻¹) and SSB (4 μM) are introduced and incubated at 37 °C for 30 min prior to EMSA examinations (Fig. 2c). As shown in Fig. 2d, a species (Species III) with both Cy3 and Cy5 signals migrating faster than the unbound pTF1-Cy5–DNA is synthesized, indicating successful strand elongation from the 3′ of IP$_{T1-DNS}$-Cy3 primer to the upstream end of pTF1-Cy5 fragment. The results provide clear evidence that the designed IP primer can be hybridized to the exposed non-target strand of DNA in the Cas9 ribonucleoprotein complex and initiate subsequent linear SDA reactions, thus demonstrating the feasibility of triggering subsequent exponential SDA reactions.

**CRISPR–Cas9-triggered exponential SDA reactions**. To evaluate whether CRISDA can initiate exponential SDA reactions by a pair of Cas9 ribonucleoproteins and IP primers in full length, the unlabeled pTF1 fragment is selected as a target. A pair of sgRNAs

(sg$_{pTF1-UPS/DNS}$) and IP primers (IP$_{pTF1-UPS/DNS}$) targeting the upstream (UPS) and downstream (DNS) sites of pTF1 are designed for selective amplification of a 186 bp sequence (Fig. 3a). Briefly, pTF1 targets at different concentrations are prepared in a diluted buffer containing 100 ng μL⁻¹ of pEGFP plasmid and 0.5 mg mL⁻¹ BSA as the background. Afterwards, CRISDA components A (Cas9 ribonucleoproteins) and B (IP primer pair and enzyme mixtures) are sequentially added to the target solution to assemble a 20 μL reaction system, which is incubated for 90 min at 37 °C prior to analyses by 6% native PAGE. As shown in Fig. 3b (upper graph), a specific amplification product is clearly observed from the reaction containing pTF1 target down to 25 aM and sequencing results confirm that it matches the target sequences. On the other hand, in the absence of Cas9, no amplification is observed under similar target concentrations (Supplementary Fig. 1) furnishing proof that Cas9 as an essential component in the CRISDA reaction. It is noticed that as the concentration of pTF1 target is lowered, the intensity corresponding to CRISDA-specific product decreases, while non-specific products with lower molecular weight derived

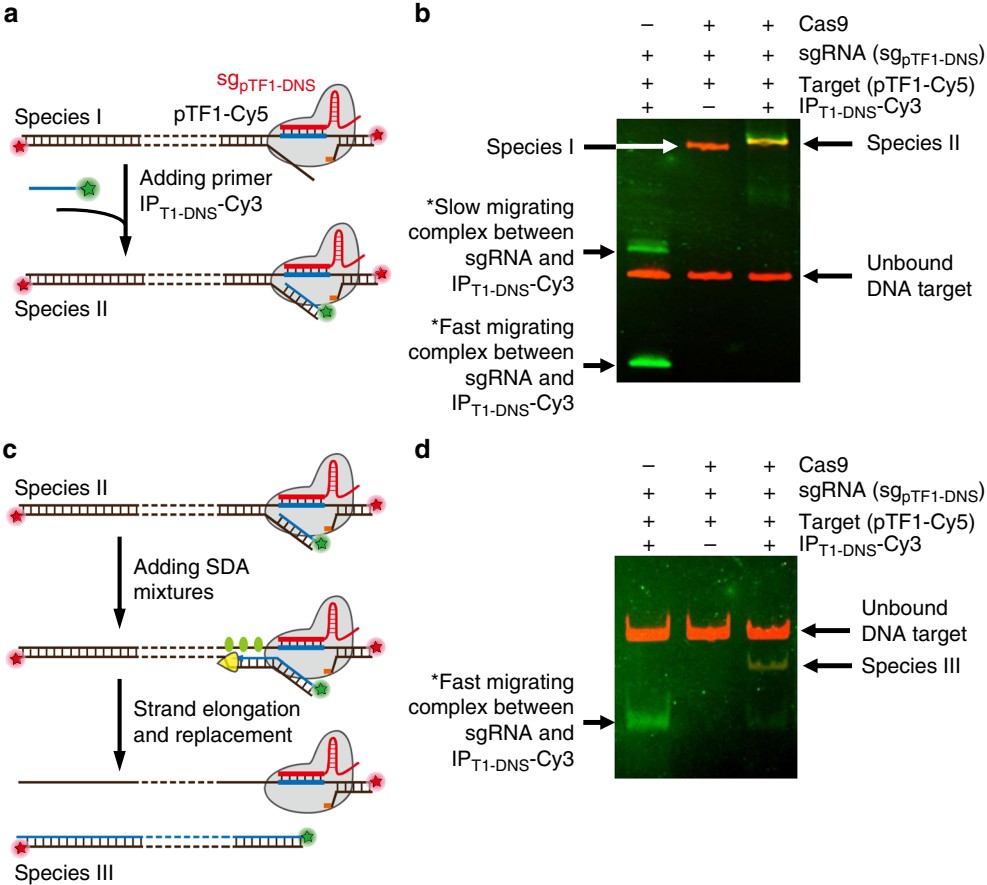

**Fig. 2** Schematic representation and fluorescent EMSA verification of CRISPR–Cas9-triggered linear SDA. **a** Schematic illustration showing binding of Cy3-labeled IP primer (green) to the exposed region of the non-target strand in the Cy5-labeled DNA–Cas9 ribonucleoprotein complex (red). **b** Fluorescent EMSA (6% PAGE) revealing the formation of DNA–Cas9 ribonucleoprotein complex (Species I) and IP–DNA–Cas9 complex (Species II). **c** Schematic illustration showing initiation of linear SDA from the IP primer after adding the SDA mixtures. **d** Fluorescent EMSA (6% PAGE) confirming successful strand elongation from the 3′ of IP$_{T1-DNS}$-Cy3 primer to the upstream end of the pTF1-Cy5 fragment. * The slow migrating complex represents the DNA–RNA hybrid formed between IP$_{T1-DNS-Cy3}$ and partially unfolded sg$_{pTF1-DNS}$, whereas the fast migrating complex represents the hybrid between IP$_{T1-DNS-Cy3}$ and fully folded sg$_{pTF1-DNS}$. Uncropped gels are shown in Supplementary Fig. 14

from primer dimers increase. This phenomenon has been observed in various nucleic acid amplification techniques such as traditional PCR reactions, HAD, and LAMP[6,32,33]. Hence, to quantitatively distinguish the target-specific product from non-target-specific products, a PNA invasion-mediated endpoint measurement is performed by incubating 20 μL of the amplification products with a biotin-labeled PNA and a Cy5-labeled PNA probe (both at 100 nM) targeting the middle region of the amplicon, which is subject to magnetic pull-down and fluorescence measurements (Fig. 1, Stage 5). In combination with this PNA invasion-mediated approach, CRISDA successfully detects pTF1 targets down to 2.5 aM (30 copies per reaction, Supplementary Fig. 2) and 0.25 aM (3 copies per reaction, Fig. 3b, lower graph) via PAGE analysis and fluorescence spectroscopy, respectively. Further analysis reveals that the CRISDA results determined by fluorescence spectroscopy exhibit a strong linear correlation between 0.25 aM and 25 fM (Supplementary Fig. 3). Moreover, successful target amplifications are accomplished when the CRISDA reactions are carried out between 25 and 43 °C and the optimal temperature is 37 °C (Supplementary Fig. 4). Thus, the subsequent CRISDA reactions are conducted at 37 °C and quantitatively monitored by PNA invasion-mediated fluorescence spectroscopy.

It is found that the 3′ overhang in IP primer complementary to the double-stranded region of the non-target strand is necessary and the melting temperature must be above 50 °C to trigger the

exponential SDA reactions (Supplementary Fig. 5). This phenomenon is caused by two reasons. First of all, this indicates that KF polymerase cannot replace the Cas9–sgRNA complex from the binding site during strand elongation and the linear SDA reactions are terminated at the border of opposite Cas9 binding site, reflecting the ultra-stability of the DNA–Cas9–sgRNA complex[34]. If elongation of the new strand from IP primer can replace the Cas9–sgRNA complex bound at the opposite site, an IP primer without the 3′ overhang should be sufficient to trigger subsequent exponential SDA reactions. Thus, the 3′ overhang in IP primer annealed to the linearly replaced strand is essential to the initiation of subsequent exponential SDA reactions (Fig. 1, Step 4). Second, although CRISDA reactions are carried out at 37 °C, the presence of SSB lowers the actual melting temperature dsDNA[35,36] and thus a long 3′ overhang in IP primer with a theoretic melting temperature over 50 °C is required for efficient annealing.

**Ultrasensitive DNA detection by CRISDA.** The sensitivity and versatility of CRISDA in detecting biologically relevant DNA targets are assessed. An 877 bp DNA fragment derived from Chromosome 9 in the human genome (Fragmented Target 1 from human genome, namely hTF1) is used as a target. Two pairs of sgRNAs (sg$_{hTF1-UPS1/DNS1}$ and sg$_{hTF1-UPS2/DNS2}$) and IP primers (IP$_{hTF1-UPS1/DNS1}$ and IP$_{hTF1-UPS2/DNS2}$) are designed to

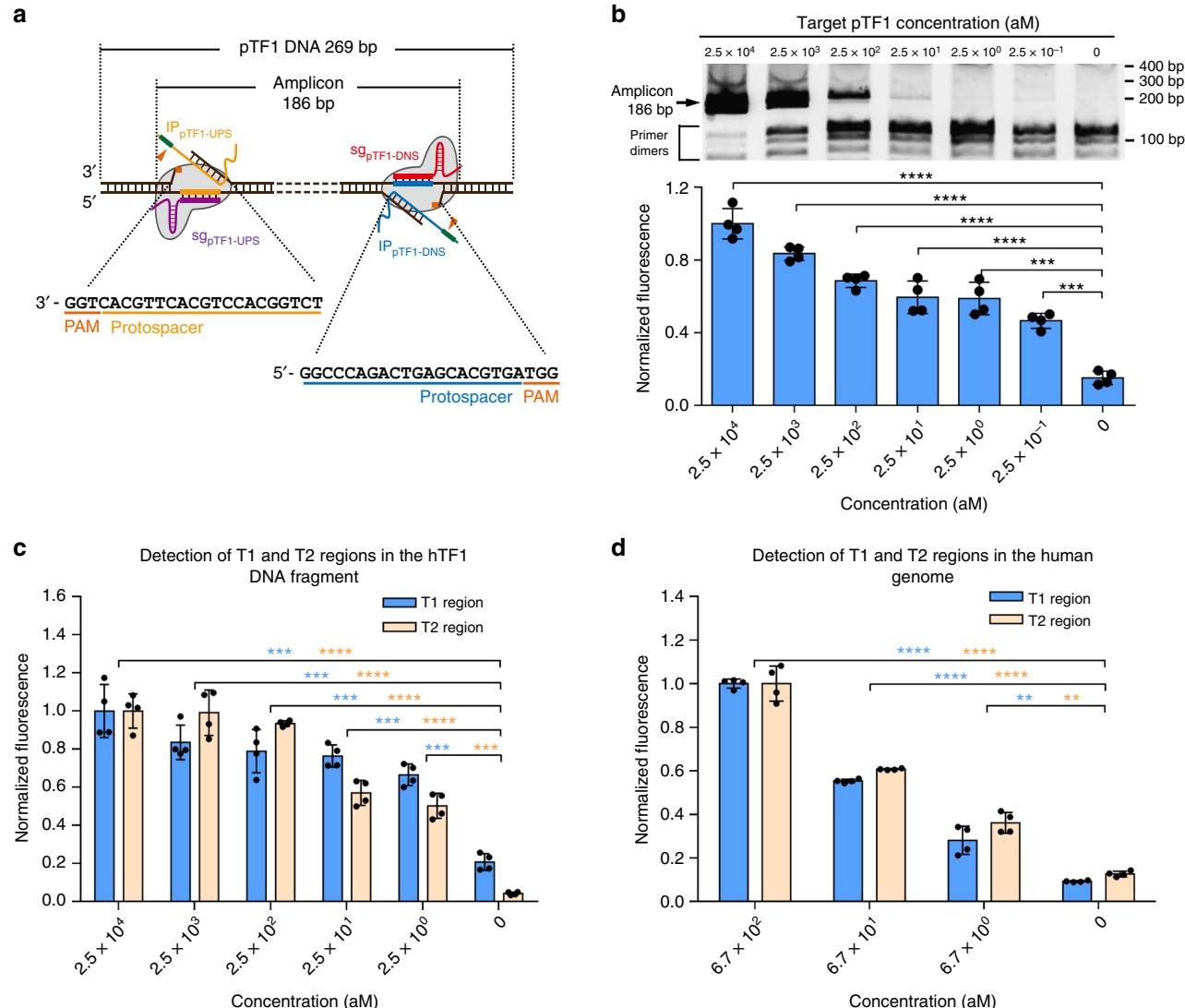

**Fig. 3** CRISDA-based DNA amplification and detection with attomolar sensitivity. **a** Schematic of CRISDA-based DNA amplification and detection towards a 269 bp model DNA fragment pTF1. **b** Representative gel and endpoint fluorescence measurements illustrating that CRISDA is capable of highly sensitive amplification and detection of the DNA fragment pTF1. The arrow indicates successful amplification of a 186 bp amplicon. **c** Representative endpoint fluorescence measurements showing that CRISDA is capable of highly sensitive amplification and detection of a 169 bp (T1) and 203 bp (T2) region in the hTF1 DNA fragment derived from Chromosome 9 in the human genome. **d** Representative endpoint fluorescence measurements showing that CRISDA is capable of highly sensitive and specific amplification and detection of the T1 and T2 region in human genomic DNA samples. In each replicate, fluorescence intensities of CRISDA reactions with various target concentrations were normalized against the one containing the highest target concentration. $n = 4$ technical replicates, two-tailed Student's $t$ test; **$P < 0.01$, ***$P < 0.001$, ****$P < 0.0001$, bars represent mean ± s.d. Uncropped gels are shown in Supplementary Fig. 14

specifically target and amplify a 169 bp (T1) and 203 bp (T2) region in hTF1, respectively (Supplementary Fig. 6). In combination with PNA, a detection limit of 2.5 aM towards both T1 and T2 sites is observed and the superior versatility of the CRISDA technique is demonstrated (Fig. 3c).

We then examine whether CRISDA is effective in detecting a specific region in the human genome and it is more challenging than detecting short DNA fragments and microbial/viral genomes because of the larger size and higher genetic complexity of the human genome. Since isolated genomic DNA contains long regions of ssDNA[37], fourfolds of SSB (16 μM) are used for genomic CRISDA amplification. CRISDA targeting T1 and T2 regions in the human genome both show a detection limit of 6.7 aM, corresponding to 0.5 ng and 83 copies of human genomic DNA per 20 μL reaction (Fig. 3d). Parallel native PAGE and sequencing analyses also confirm successful amplification of target regions but in contrast, no

amplification is detected from the reactions without Cas9 (Supplementary Fig. 7). Although a detection range between 2.5 aM and 25 fM has been investigated for the target fragment hTF1 (877 bp), the maximum concentration of genomic DNA in CRISDA amplification is 670 aM, corresponding to 50 ng genomic DNA in every 20 μL CRISDA reaction. Owing to the large size of human genomic DNA ($6 \times 10^9$ bp and ~6.1 pg per genome), further increase of the human genomic DNA to 500 ng (6.7 fM) or 5 μg (67 fM) in every 20 μL reaction will be too high for most realistic applications. Therefore, a detection range of 6 and 4 orders of magnitude (including 0 as a negative control) is used in CRISDA amplification towards fragment DNA targets and genomic DNA, respectively.

Owing to the increasing diversity of genetically modified organism (GMO) products sold in the market, cost-effective, and programmable detection methods with high sensitivity and specificity are highly desirable. Here, we further evaluate if

CRISDA can detect low-abundance of GMO contents in the high-levels of wild-type counterparts. A genetically modified soybean, MON87705, is chosen as a target and a pair of sgRNAs (sg$_{gTF1-UPS/DNS}$) and IP primers (IP$_{gTF1-UPS/DNS}$) are designed to specifically target and amplify a 194 bp region at the boundary of the wild-type genome and transgenic insert region (Fragmented Target 1 from GMO soybean, namely gTF1, 929 bp, Fig. 4a). Genomic DNA from the wild-type soybean is used as the background to evaluate the specificity of CRISDA. In the presence of 75 ng $\mu L^{-1}$ wild-type soybean genomic DNA, CRISDA detects 2.5 aM and 25 aM gTF1 fragment using fluorescence spectroscopy and PAGE analysis, respectively (Fig. 4b, and Supplementary Fig. 8). In addition, genomic CRISDA also detects 25 pg $\mu L^{-1}$ MON87705 genomic DNA (corresponding to 36.6 aM, and 439 copies per 20 $\mu L$ reaction) in the presence of 75 ng $\mu L^{-1}$ wild-type soybean genomic DNA as the background (Fig. 4c). In comparison, traditional PCR approaches are used to amplify the GMO fragment gTF1 and GMO genomic DNA and the PCR products are subsequently analyzed by PAGE and PNA invasion-mediated endpoint measurements. As shown in Supplementary Figs. 9a, b, the designed GMO-For/Rev primer pair successfully amplifies 1 ng (25 pM) to 0.1 pg (2.5 fM) of target gTF1 and 50 ng (3.66 fM) of GMO genomic DNA diluted without background, confirming that the GMO-For/Rev primer pair is effective in standard PCR. However, in the presence of interfering DNA and BSA as the background, PCR fails to produce detectable amplicons below 25 fM gTF1 and 3.66 fM GMO genomic DNA as templates (Supplementary Figs. 9c, d). In addition, only weak fluorescent signals are observed by the PNA invasion-mediated method from the PCR products containing 25 and 2.5 fM gTF1 as templates (Supplementary Figs. 9e, f). The results indicate that the sensitivity of CRISDA is at least three orders of magnitude higher than that of traditional PCR under the same conditions.

**Single-nucleotide specific DNA detection by CRISDA.** On account of the low mismatch tolerance of Cas9 ribonucleoprotein in recognizing the target DNA, especially in the PAM and seed (PAM-proximal) sequences, the capability of CRISDA in distinguishing single-nucleotide mismatch is investigated. Fragments of pTF1 mutants bearing different single-nucleotide mutations in the PAM and seed sequences at the DNS site are constructed as

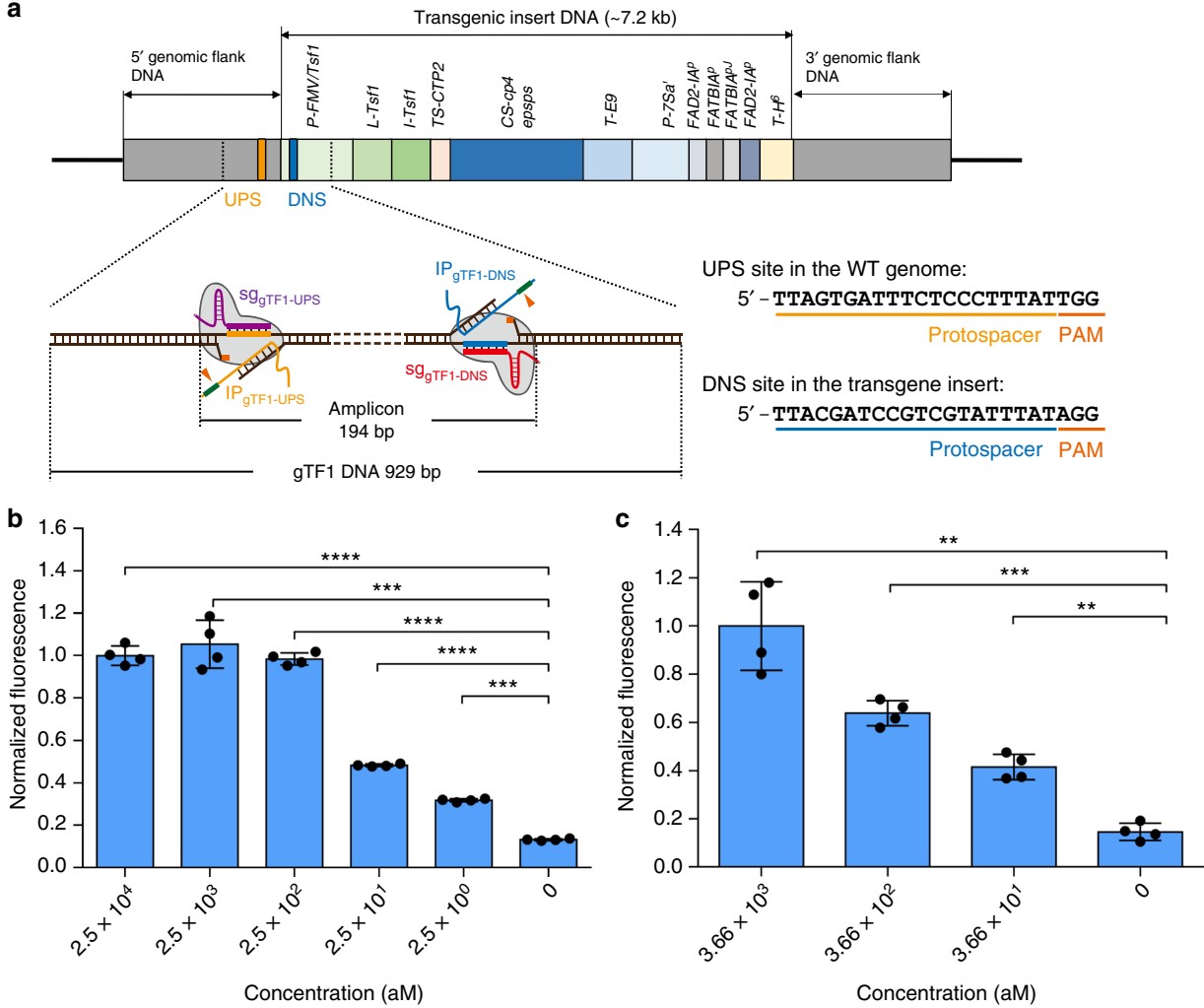

**Fig. 4** CRISDA-based detection of low-abundance of GMO contents in the high-levels of wild-type counterparts. **a** Schematic showing that CRISDA specifically target and amplify a 194 bp region at the boundary of the wild-type genome and transgenic insert region in the genome of the genetically modified soybean, MON87705. Representative endpoint fluorescence measurements showing that **b** CRISDA can sensitively detect a PCR-amplified GMO fragment (gTF1, 929 bp) and **c** CRISDA can sensitively and specifically detect MON87705 genomic DNA in the presence of wild-type soybean genomic DNA. In each replicate, fluorescence intensities of CRISDA reactions with various target concentrations were normalized against the one containing the highest target concentration. $n = 4$ technical replicates, two-tailed Student's $t$ test; **$P < 0.01$, ***$P < 0.001$, ****$P < 0.0001$, bars represent mean ± s.d

templates (Fig. 5a) and sg_pTF1-UPS/DNS and IP_pTF1-UPS/DNS pairs targeting wild-type UPS and DNS site are used in the CRISDA reactions. According to native PAGE analyses and PNA invasion-mediated fluorescence measurements, significant differences are observed between the wild-type template and templates with single-nucleotide mutations in the PAM and the first nucleotide in the seed region, while the difference in the second nucleotide mutant is not statistically significant (Fig. 5b). In contrary, single-nucleotide mutations localized between the third and fifth nucleotides in the seed region, which are previously regarded as a critical region in determining the overall specificity of Cas9 ribonucleoprotein, exhibit weak effects in the CRISDA performance (Supplementary Fig. 10). This phenomenon stems from the characteristics of Cas9 ribonucleoproteins. Since various genome-wide studies have revealed off-target effects at sites homologous to the on-target site with only single-nucleotide mutations in the PAM-proximal region[23,38], we propose that a minor population of mutant templates can still be recognized and nicked by Cas9–sg_pTF1-DNS complex in CRISDA reactions and exponential amplifications are subsequently initiated. On the

other hand, mutants with single-nucleotide mismatch at the PAM and the first nucleotide can fully block the recognition by Cas9 to prevent subsequent amplification. The results confirm that the Cas9 ribonucleoprotein plays a key role in the sensitivity and specificity of CRISDA.

Based on the ability to distinguish single-nucleotide discrepancy, CRISDA is further adopted in disease-related heterozygous SNPs genotyping. A locus containing a breast cancer-associated SNP (rs3803662, CCT/TTC) is chosen as a target[39,40] and genomic DNA is extracted from human embryonic kidney 293 cells (HEK293) and human breast cancer cells (MCF7 and T47D). The fragments bearing rs3803662 site (Fragmented Target 2 from human genome, namely hTF2) are subjected to PCR amplification and sequencing verification (Fig. 5c). A pair of sgRNAs (sg_hTF2-UPS/DNS) and IP primers (IP_hTF2-UPS/DNS) are designed for selective amplification of a 186 bp sequence in the hTF2 fragment with the rs3803662 wild-type sequences CCT serving as the PAM at the downstream site. As shown in Fig. 5d and Supplementary Fig. 11, CRISDA successfully discriminates down to 25 aM fragments with high significance inferring both

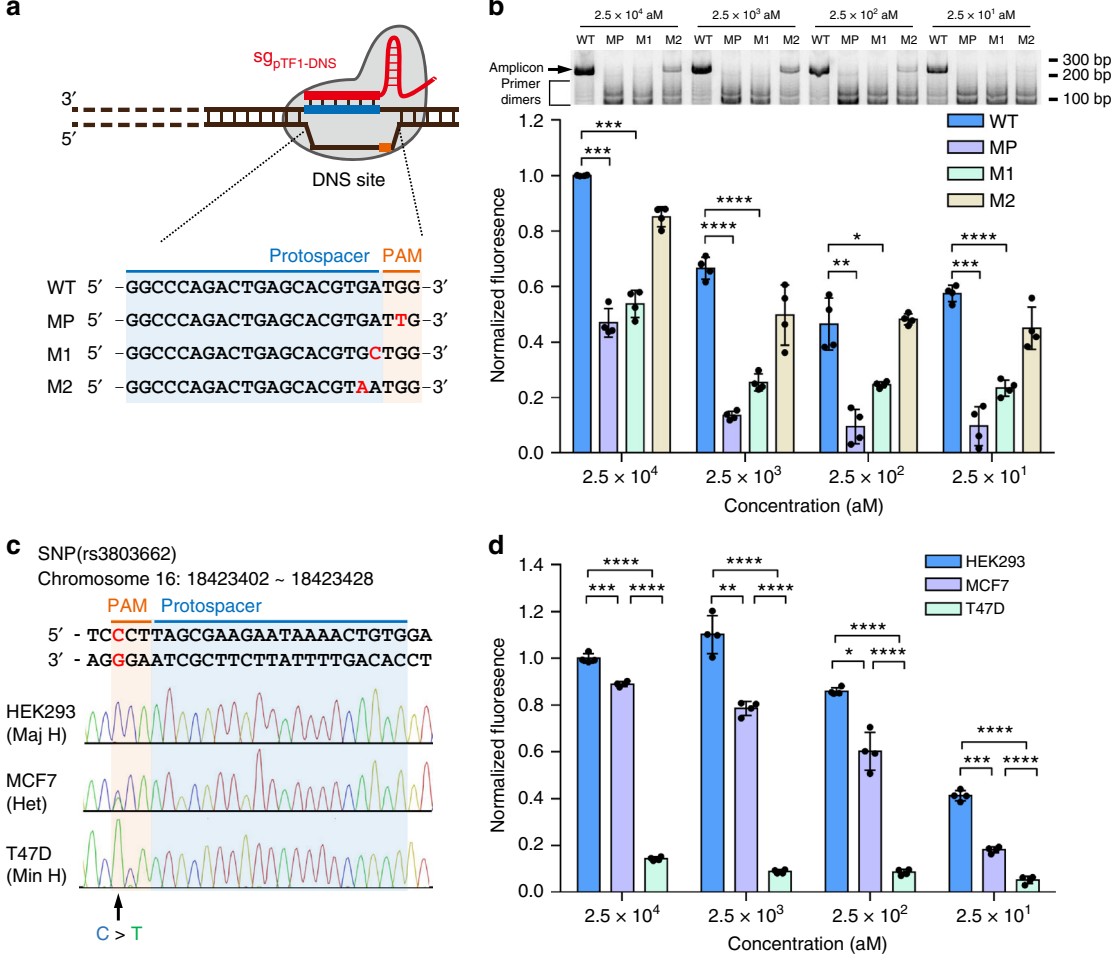

**Fig. 5** CRISDA-based DNA amplification and detection with single-nucleotide specificity. **a** Schematic of wild-type pTF1 fragment and various mutants bearing single-nucleotide mutations in the PAM and seed sequences at the DNS site. WT: wild type, MP: PAM mutant, M1: +1 mutant, and M2: +2 mutant. **b** Representative PAGE analyses and endpoint fluorescence measurements confirming that CRISDA can specifically discriminate pTF1 fragments with single-nucleotide mutations at the PAM and +1 sites in the seed sequence. **c** Sequencing verification of a breast cancer-associated SNP (rs3803662, CCT/TTC) in the genomes from HEK293 (major homozygotes, Maj H, CC), MCF7 (heterozygotes, Het, CT), and T47D (minor homozygotes, Min H, TT) cells. **d** Representative endpoint fluorescence measurements showing that CRISDA can discriminate SNP for highly sensitive human genotyping. In each replicate, fluorescence intensities of CRISDA reactions with various target concentrations were normalized against the one containing the highest target concentration. $n = 4$ technical replicates, two-tailed Student's $t$ test; $*P < 0.05$, $**P < 0.01$, $***P < 0.001$, $****P < 0.0001$, bars represent mean ± s.d Uncropped gels are shown in Supplementary Fig. 14

homozygous and heterozygous genotypes. Specifically, the strongest fluorescence signals are observed from the CRISDA reactions using DNA fragments from HEK293 genome as templates, where the rs3803662 site is CCT (the major homozygotes with low-risk of breast cancer susceptibility). Meanwhile, the CRISDA reactions using fragments from the genomes of MCF7 as the template, which is a heterozygote allele carrier at the rs3803662 site with medium-risk of breast cancer, generate statistically lower signals compared to the wild type. Interestingly, the results of the hTF2 fragments from T47D cells, the minor homozygotes with CTT at the rs3803662 site (high-risk of breast cancer susceptibility), reveal undetectable fluorescent signals even at a high concentration. The results clearly demonstrate the outstanding sensitivity and specificity of CRISDA boding well for therapeutic detection of disease-related SNPs.

**CRISDA combined with targeted enrichment of DNA by Cas9.**
Targeted enrichment of DNA is widely implemented in biomedical fields including pathogen detection, genetic diagnosis, and next-generation sequencing[41,42]. Since Cas9-mediated specific enrichment of target DNA and RNA has been reported previously[43,44], we explore if CRISDA can be integrated with Cas9-mediated DNA enrichment to further enhance the sensitivity in DNA detection. Briefly, N-terminal biotin-labeled dCas9 ribonucleoprotein is incubated with 1 mL of PBS buffer containing 2.5 aM target DNA fragment (hTF1) for 15 min in the presence of 100 ng $\mu L^{-1}$ plasmid DNA and 0.5 mg $mL^{-1}$ BSA as the background (Fig. 6a; Construction of biotin-labeled dCas9 and its binding activity are described in Supplementary Fig. 12). After addition of streptavidin-labeled magnetic beads and subsequent magnetic pull-down, the standard CRISDA reactions are performed without further washing or elution. As confirmed by PNA-mediated fluorescence measurements and native PAGE analyses, CRISDA exhibits at least 100-folds enhanced sensitivity in combination with Cas9-mediated enrichment of DNA (Fig. 6b, and Supplementary Fig. 13). It clearly demonstrates the great potential of CRISDA in rapid and robust detection of target DNA molecules with sub-attomolar sensitivity.

## Discussion
The results described in previous sections demonstrate that CRISDA is a true isothermal DNA amplification and detection technique with attomolar sensitivity and single-nucleotide

specificity and it is achieved by combining the sensitive and specific recognition of CRISPR effector Cas9 towards its target DNA and efficient exponential amplification of SDA. Besides the high sensitivity and specificity in a complex background, this versatile and robust technique possesses distinctive advantages over traditional PCR-based methods and most isothermal DNA amplification techniques. First, CRISDA is a true isothermal reaction that can be performed at one temperature for the entire process. In the CRISDA reactions, unwinding of target duplex DNA at primer binding sites is facilitated by Cas9-targeted recognition and cleavage[25,26], thus eliminating the requirements for expensive thermocycler in PCR-based methods or the initial pre-treatment to expose ssDNAs in most isothermal amplification techniques such as the padlock/MIP methods, conventional SDA, and LAMP[2,4,5]. Second, the primer design is relatively simple compared to other isothermal techniques. In the CRISDA reactions, only one pair of IP primer is required to trigger the linear strand replacement reaction and subsequent exponential amplification. In addition, each IP primer is simply composed of a 5′ Nb.BbvCI endonuclease nicking site, middle hybridization region, and 3′ overhang complementary to unwound region of the nontarget strand. Our results demonstrate that as long as the melting temperature of 3′ overhang is above 50 °C, the IP primers can efficiently initiate exponential amplification without the necessity of further optimization. On the other hand, the other two components, sgRNAs and PNA probes, both have limited effects in the overall performance of the CRISDA reactions (details are described in the Methods section), thereby making CRISDA a pragmatic technique in the detection of new DNA targets. In contrast, multiple primer pairs or long probes are required and subjected to further optimization by other isothermal amplification techniques such as the conventional SDA, LAMP, and padlock/MIP-mediated methods[4,5,45]. Third, CRISDA can be extended to meet different requirements in nucleic acids detection. Since spyCas9 has been utilized in targeted mRNA pull-down[43], CRISDA can be easily applied to RNA amplification and detection by including a reverse transcription step. Additionally, a diverse set of CRISPR effectors with different properties exist in nature and can provide opportunities to further improve the CRISDA effectiveness and performance. For example, AceCas9 (from *Acidothermus cellulolyticus*) recognizes a 26 bp sequence[46] rather than the 20 bp in spyCas9 and may result in more efficient IP primer binding and SDA initiation. On the other hand, Cas9 variants with different PAM requirements provide a broad range of choices in SNPs detection by CRISDA[47].

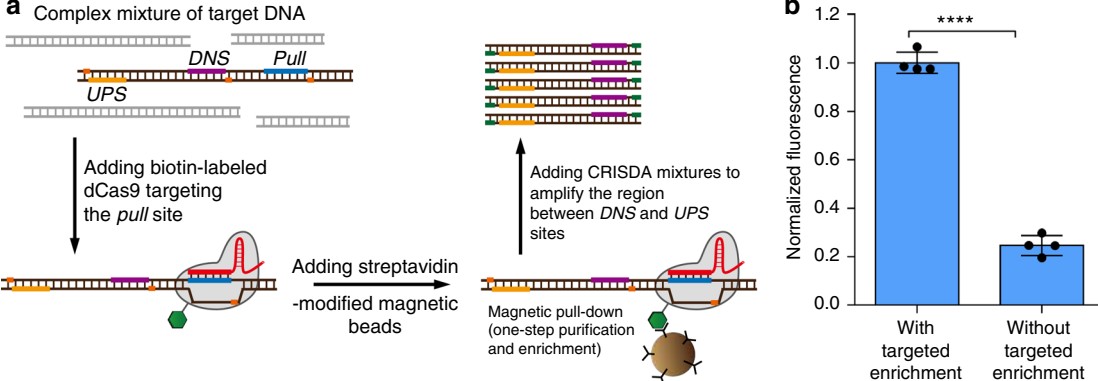

**Fig. 6** Targeted enrichment of DNA by Cas9 further enhances the sensitivity of CRISDA. **a** Schematic of the Cas9-mediated specific enrichment of target DNA in combination with CRISDA. **b** Representative endpoint fluorescence measurements showing that the sensitivity and reliability of CRISDA are significantly enhanced when combined with Cas9-mediated enrichment of target DNA (reactions are conducted at an hTF1 concentration of 2.5 aM). In each replicate, fluorescence intensity of the CRISDA reaction without targeted enrichment were normalized against the one with targeted enrichment. $n =$ 4 technical replicates, two-tailed Student's $t$ test; ****$P < 0.0001$, bars represent mean ± s.d

Although spyCas9-mediated CRISDA has great sensitivity and specificity in DNA detection, two issues need to be optimized further in future. First of all, CRISDA exhibits varied dynamic ranges and amplification efficiencies towards different types of targets (target length, sequence complexity and GC-richness). This phenomenon also prevails in PCR-based methods, as it is well-known that long genomic, complex, and GC-rich templates are difficult to amplify[48–50]. Further experiments are required to optimize the reaction conditions and to choose DNA polymerases with stronger strand displacement activity and higher processivity than KF polymerases. Second, non-specific products from primer dimers are observed especially when the target level is very low. Since the primer dimer is normally generated at a low temperature, further optimization of CRISDA should be carried out using Cas9 ribonucleoproteins working at a higher temperature, for example, AceCas9 and GeoCas9 (from *Geobacillus stearothermophilus*)[46,51]. In combination with the thermostable CRISPR effectors, nickases, and DNA polymerases with strong strand displacement activity and high processivity such as the large fragment of *Bst* DNA polymerase, the specificity, sensitivity and efficiency of CRISDA can be improved further to accomplish real-time measurements.

In summary, CRISDA is demonstrated to be a true isothermal approach in nucleic acid detection with attomolar sensitivity and single-base specificity in the presence of a complex sample background. This versatile and robust method has great potential in energy-efficient portable diagnostic devices as well as applications such as point-of-care diagnostics and field analyses.

## Methods

**Materials**. The pET28a/Cas9-Cys expression vector was purchased from Addgene (Addgene #53261). The engineered plasmids modified from pET28a/Cas9-Cys vector, such as pET28a/Cas9(H840A)-Cys (plasmid expressing *S. pyogenes* Cas9 protein with a mutation of HNH catalytic residue) and pET28a/S3C-dCas9 (plasmid expressing *S. pyogenes* Cas9 protein with mutations of RuvC and HNH catalytic residues, and bearing only one Cysteine as the 3rd amino acid), were constructed by GenSript Biotechnology and verified by sequencing. The sequences of the Cas9 coding regions in plasmid pET28a/Cas9(H840A)-Cys and pET28a/S3C-dCas9 are listed in Supplementary Tables 1 and 2, respectively. Oligonucleotides used in this study were synthesized by IGE Biotechnology and the sequences of oligonucleotides are listed in Supplementary Tables 3 to 8. PNAs for strand invasion and fluorescence detection were synthesized by and Tahepna Biotechnologies. All the PNA sequences are listed in Supplementary Table 9. Platinum™ *Taq* DNA polymerase, EZ-Link® Maleimide-PEG2-Biotin and magnetic beads coated with streptavidin (Dynabeads™ MyOne™ Streptavidin C1) were purchased from ThermoFisher. Proteins and other reagents such as single-strand binding protein (T4 gene 32 protein), Nb.BbvCI endonuclease, DNA polymerases (Klenow Fragment 3′–>5′ exo−, KF), and dNTPs used in CRISDA amplification reactions were obtained from New England Biolabs (NEB). Bovine serum albumin (BSA, 20 mg mL$^{-1}$), and nuclease free water were purchased from TAKARA. The other chemicals used in this study were purchased from Sigma-Aldrich in analytical reagent grade and used without further purification.

**Expression and purification of Cas9 proteins**. Engineered Cas9 expression vectors, pET28a/Cas9(H840A)-Cys, and pET28a/S3C-dCas9, were used for the expression of Cas9(H840A) and S3C-dCas9 proteins, respectively. Briefly, the plasmids were transformed into *E. coli*. BL21(DE3) (Stratagene) competent cells. The transformed cells were cultured in 250 mL LB medium overnight at 37 °C and 0.1 mM IPTG was added to induce protein expression for another 14–16 h at 23 °C. Afterwards, the cells were harvested by centrifugation (6000×*g*, 4 °C, 15 min) and resuspended in 50 mL of lysis buffer (20 mM Tris-HCl, 300 mM NaCl, 20 mM imidazole, and 0.1% Tween 20, pH 8.0). After sonication for cell lysis and centrifugation at 9400×*g* for 15 min at 4 °C, the supernatant was transferred to Poly-Prep chromatography columns (Bio-Rad), and incubated with 500 μL Ni-NTA agarose beads (QIAGEN) for 1–2 h at 4 °C. After two rounds of washing by 10 mL of lysis buffer, the bound proteins were eluted by 1 mL of elution buffer (20 mM Tris-HCl, 300 mM NaCl, 300 mM imidazole, and 0.1% Tween 20, pH 8.0). The eluted proteins were dialyzed overnight in dialysis buffer consisting of 20 mM Tris-HCl, 250 mM KCl, 0.5 mM TCEP, 0.5 mM EDTA, and 0.1% Tween 20, pH 8.8. The purified proteins were analyzed by SDS-PAGE and quantified by Bradford assays (Bio-Rad) and finally, the aliquoted proteins were subject to snap-freeze in liquid nitrogen and stored at −80 °C.

**sgRNA design and synthesis**. The sgRNAs were in vitro transcribed from the T7 promoter driven sgRNA (single guide RNA) expression vector pDR274 (Addgene #42250). Briefly, the BsaI-digested pRD274 vector was ligated with the annealed oligonucleotide harboring a customized 20 bp targeting sequence and sub-cloned in *E. coli* DH5α. The constructed plasmids were extracted, and digested by restriction endonuclease DraI. The digested products (290 bp) were subject to gel extraction, and subsequently 100 ng purified DNA fragments were transcribed into sgRNAs in vitro using high yield MEGAscript™ T7 transcription kit (ThermoFisher) according to the standard protocol. The transcribed sgRNAs were ethanol precipitated and quantified by spectrometry. The oligonucleotides designed for the construction of variant pDR274-sgRNA expression plasmids are listed in Supplementary Table 5.

**Construction of vectors containing pTF1 and its mutants**. pGL3-basic vector (Promega) was first digested by SacI and NheI restriction endonucleases, followed by a gel extraction of the 4804 bp plasmid backbone. Then, a 100 bp PCR-amplified Luciferase coding region (primer pair: pGL3-100-For/Rev) was ligated with the purified plasmid backbone giving the vector pGL3-100. The gel extracted pGL3-100 vector was subject to the second round of double digestion using XhoI and HindIII restriction endonucleases. After gel extraction, the digested plasmid backbone was ligated with the annealed oligonucleotide harboring the wild type or single-nucleotide mutated 20 bp protospacer and PAM sequences (the DNS site) giving pGL3-100-Target$_{WT}$ and its mutant vectors (pGL3-100-Target$_{MP/M1~M5}$). All the oligonucleotides used for the construction of pTF1 and the mutant fragments are listed in Supplementary Table 4.

**DNA extraction**. The human embryonic kidney cells 293 (HEK293, Cat. No.: 3142C0001000001636), and human breast cancer cells (MCF7, Cat. No.: 3142C0001000000054 and T47D, Cat. No.: 3142C0001000000045) were all purchased from China Type Culture Collection (CTCC). The human genomic DNAs were extracted from these cells with Tissue DNA kit (Omega) according to the manufacture's instruction. Sequencing verification of the breast cancer-associated SNP (rs3803662, CCT/TTC) in the genomes from HEK293 (major homozygotes, Maj H, CC), MCF7 (heterozygotes, Het, CT) and T47D (minor homozygotes, Min H, TT) cells was performed using primer pair hTF2-For/Rev to authenticate cell lines. The genomic DNAs from the Certified Reference Materials (CRMs) of genetically modified (GM) soybean MON87705 (AOCS, 0210-A) and non-modified soybeans (AOCS, 0911-A) were used as a positive sample and background interferences, respectively. The soybean genomic DNAs were extracted from the devitalized seed powder of homozygous MON87705 soybeans and non-modified soybeans using HP Plant DNA kit (Omega). The plasmid pEGFP as background DNA was extracted by QIAGEN® Plasmid Midi Kit (QIAGEN).

**Preparation of target fragments for CRISDA reactions**. The Cy5-labeled template pTF1-Cy5 was amplified by PCR from pGL3-100-Target$_{WT}$ plasmid using 5' Cy5-labeled primer pair RV3$_{Cy5}$ and GL2$_{Cy5}$. The unlabeled pTF1 fragment and its mutated fragments bearing different single-nucleotide mutations at the DNS site were amplified by PCR from corresponding pGL3-100-Target$_{WT/MP/M1~M5}$ plasmids (primer pair: pTF1-For/Rev). The hTF1 fragment was PCR amplified from the chromosome 9 in the human genome using the primer pair hTF1$_{-For/Rev}$. The hTF2 fragment was PCR amplified from the chromosome 16 in the human genome bearing rs3803662 SNP site (primer pair: hTF2-For/Rev). The gTF1 fragment was obtained from the genome of GM soybean M0N87705 by genomic PCR amplification using primer pair gTF1-For/Rev. All the PCR amplifications were carried out using Platinum™ *Taq* DNA polymerase following the manufacturer's instructions. All PCR-amplified target fragments were verified by sequencing, purified through ethanol precipitation, and quantified by spectrometry. The weight concentrations of target DNA fragments were converted to molar concentrations based on the sequence-specific molecular weights. Tenfold serial dilutions were performed to prepare target fragments solutions ranging from 2.5 μM to 0.25 aM for subsequent CRISDA reactions in dilution buffer (10 mM Tris-HCl, 50 mM NaCl, 10 mM MgCl$_2$, 0.1% Tween 20, 0.2 mg mL$^{-1}$ BSA, and 100 ng μL$^{-1}$ pEGFP, pH 8.0).

**Binding of the IP primer to the DNA–Cas9–sgRNA complex**. The Cas9–sgRNA ribonucleoprotein was constructed by incubating 100 nM sg$_{pTF1-DNS}$ and 50 nM purified Cas9(H840A) in 20 μL reaction buffer (10 mM Tris-HCl, 50 mM NaCl, 10 mM MgCl$_2$, and 0.1% Tween 20, pH 8.0) for 15 min at 37 °C. Afterwards, 1 μL of the Cy5-labeled DNA fragment, pTF1-Cy5 (2 μM) was added to the reaction and incubated for 15 min under the same temperature to form stable DNA–Cas9–sgRNA ribonucleoprotein complex. Afterwards, 0.5 μL of Cy3-labeled IP primer (IP$_{T1-DNS}$-Cy3, 2 μM) were added and incubated for 10 min at 37 °C. Finally, after addition of 5 μL 6 × native loading buffer (Takara), 10 μL of the reaction mixture was loaded to 6% native PAGE and electrophoresis mobility shift assays (EMSA) were performed. The gels were visualized and analyzed by the Tanon 6100c Chemiluminescent Imaging System with its built-in Cy3 and Cy5 channels.

**CRISPR-Cas9-triggered linear SDA**. To investigate the initiation of linear SDA from the IP primer ($IP_{T1-DNS}$-Cy3), 10 μL of the reaction mixture containing Cas9–sgRNA ribonucleoprotein complex, pTF1-Cy5 and $IP_{T1-DNS}$-Cy3 was further mixed with 10 μL of SDA mixture (10 mM Tris-HCl, 50 mM NaCl, 10 mM MgCl$_2$, 0.1% Tween 20, 0.8 U μL$^{-1}$ KF polymerase, 4 μM SSB, 0.5 mM dNTPs, and 0.4 mg mL$^{-1}$ BSA, pH 8.0). The mixture was further incubated for 15 min at 37 °C prior to the fluorescent EMSA assays.

**CRISDA reactions**. The CRISDA reaction mixtures were prepared separately as Component A and B. Component A was prepared by incubating 250 nM Cas9 (H840A) protein and 500 nM of sgRNA in reaction buffer (10 mM Tris-HCl, 50 mM NaCl, 10 mM MgCl$_2$, and 0.1% Tween 20, pH 8.0) for 15 min at 37 °C. Component B was assembled in the reaction buffer containing IP primer pair (100 nM each), 0.4 U μL$^{-1}$ Nb.BbvCI endonuclease, 0.8 U μL$^{-1}$ KF polymerase, 4 μM SSB, 0.5 mM dNTPs, and 0.4 mg mL$^{-1}$ BSA (for genomic CRISDA reactions, concentration of SSB was 16 μM). Component A and B can be stably stored at -20 °C for months. In the typical CRISDA reaction, 9 μL of the target fragments solution with interfering DNA and BSA as backgrounds was mixed with 1 μL of Component A and incubated for 15 min allowing the formation of DNA–Cas9 ribonucleoprotein complex, and nicking/exposure of the non-target strands. Afterwards, 10 μL of Component B were added and the 20 μL of the assembled mixture were incubated for 90 min to exponentially amplify targeted amplicon. The incubation steps were performed at 37 °C (except for the experiments investigating temperature tolerance of CRISDA). After amplification, the reaction products were mixed with 4 μL 6 × native loading buffer and 1 μL of 20x Gel Green in water (Biotinum), prior to the fluorescent EMSA assays using 6% native PAGE. The gels were visualized and analyzed by the Tanon 6100c Chemiluminescent Imaging System with its built-in FAM channel.

**PNA invasion-mediated fluorescence measurements**. To quantitatively distinguish the target-specific product from the non-target-specific products in CRISDA reactions, PNA invasion-mediated fluorescence measurements were performed by incubating the CRISDA reaction with a biotin-labeled PNA and a Cy5-labeled PNA probe (both at 100 nM) targeting the middle region of the amplicon, which was subject to magnetic pull-down and fluorescence measurements. Briefly, 2 μL of the PNA mixture (biotin-labeled PNA and Cy5-labeled PNA both at 1 μM) were added to the 20 μL of CRISDA reaction. The mixture was incubated at ambient temperature or 37 °C for 15 min. Afterwards, 3 μL of streptavidin-coated magnetic beads (Dynabeads™ MyOne™ Streptavidin C1) were introduced and the mixture was further incubated at ambient temperature or 37 °C for 15 min. Subsequently, the complex containing the specific amplicon and two PNA probes was isolated from non-specific products through magnetic pull-down. The beads were resuspended in 100 μL of the reaction buffer supplemented with 0.4% SDS and incubated at room temperature for 15 min. The fluorescence intensity of the supernatants was determined on a fluorescence spectrophotometer F-4600 (Hitachi High-Technologies Corporation). In each replicate, fluorescence intensities of CRISDA reactions with various target concentrations were normalized against the one containing the highest target concentration.

**PCR-based comparison**. In comparison, traditional PCR approaches are used to amplify the GMO fragment gTF1 and GMO genomic DNA using primer pair GMO-For/Rev. Tenfold serial dilutions were performed to prepare target solutions using dilution buffer (10 mM Tris-HCl, 50 mM NaCl, 10 mM MgCl$_2$, 0.1% Tween 20, pH 8.0) or dilution buffer supplemented with interfering DNA (75 ng μL$^{-1}$ wild-type soybean genomic DNA) and BSA (0.5 mg mL$^{-1}$) as background. The PCR amplifications were carried out using Platinum$^{TM}$ Taq DNA polymerase following the manufacturer's instructions, and the PCR products are subsequently analyzed by 6% native PAGE and PNA invasion-mediated endpoint measurements in the presence of 4 μM SSB.

**Labeling of S3C-dCas9 protein with biotin**. EZ-Link® Maleimide-PEG2-Biotin (ThermoFisher) was employed to biotinylate purified S3C-dCas9 protein according to the manufacture's instruction. Briefly, 20 mM EZ-Link® Maleimide-PEG2-Biotin was prepared in moisture-free DMSO for long-term storage. In protein biotinylation, the S3C-dCas9 protein purified in Tris-HCl buffer was first changed to PBS buffer (pH 7.0) by ultrafiltration (Amicon® Microcon, 500 μL, MWCO: 100kD) and the protein concentration was adjusted to 10 μM. Then, 20-fold molar excess of the EZ-Link® Maleimide-PEG2-Biotin (20 mM in DMSO, 1 μL) was added to S3C-dCas9 protein (10 μM in PBS, 100 μL) solution, and the mixture was incubated overnight at 4 °C with gentle shaking. The reaction was quenched by 2 mM DTT and the unlabeled biotin was removed by ultrafiltration. To investigate the activity of biotin-labeled S3C-dCas9, 100 nM biotinylated S3C-dCas9 and non-biotinylated protein were separately incubated with 200 nM sgRNA (sg$_{pTF1-DNS}$) for 15 min at 37 °C to form Cas9–sgRNA ribonucleoproteins, followed by mixing with 200 nM pTF1-Cy5 fragment for 15 min at the same temperature. The samples incubated with or without 100 nM streptomycin protein (NEB) for 5 min were analyzed by EMSA assays using 1.0% agarose gel. The gels were visualized and analyzed by the Tanon 6100c Chemiluminescent Imaging System with its built-in Cy5 channel.

**Targeted enrichment of DNA and combination with CRISDA**. The hTF1 fragment was used as a model to examine if CRISDA could be integrated with Cas9-mediated DNA enrichment to further enhance its sensitivity in DNA detection. Briefly, 500 nM biotinylated S3C-dCas9 was separately incubated with 1 μM sgRNA (sg$_{hTF1-UPS1}$) for 15 min at 37 °C in 100 μL of the reaction buffer to form Cas9–sgRNA ribonucleoprotein. 100 μL of the mixture was added to 900 μL reaction buffer containing 2.5 aM hTF1 fragment as target and 100 ng μL$^{-1}$ plasmid DNA and 0.5 mg mL$^{-1}$ BSA as backgrounds. The mixture was further incubated for 20 min at 37 °C. Afterwards, 7.5 μL of Dynabeads™ MyOne™ Streptavidin C1 were introduced and incubated for another 30 min. After magnetic pull-down, the enriched targets were subject to standard CRISDA reactions using sg$_{hTF1-UPS2/DNS2}$ and IP$_{hTF1-UPS2/DNS2}$. The amplification was analyzed by fluorescent EMSA assays using 6% native PAGE and PNA invasion-mediated fluorescence measurements by following the protocol described in the previous section (PNA invasion-mediated fluorescence measurements).

**General considerations for the design of CRISDA sequences**. In the design of sgRNA pair directing recognition of Cas9 to target sites and triggering subsequent SDA reactions, 20 bp guide sequences followed by NGG as the PAM can be randomly chosen in the forward and reverse strands (100 to 250 bp from each other). In this study, randomly chosen sgRNAs with high specificity (with only 60 potential off-target sites in the human genome) or low specificity (with more than 300 potential off-target sites) all successfully triggered subsequent CRISDA reactions, indicating that sgRNA design is not a critical factor (Supplementary Table 10). In the design of the IP primer pair, the middle region hybridized to the exposed non-target strand in the DNA–Cas9–sgRNA complex had limited effects on the performances of the CRISDA reaction. As summarized in Supplementary Table 11, the GC contents of this region in IP primers ranged from 68.8% (GC rich) to 31.2% (AT rich), with corresponding melting temperatures between 56.2 and 39.8 °C. The only critical factor in the IP primer was the melting temperature of the 3′ overhang complementary to the double-stranded region of the non-target strand, which has been discussed in details. In the design of PNA probes targeting the middle region of amplicons, two simple rules should be adopted. First, the melting temperature of the PNA probe with target DNA must be over 37 °C. Second, because of the strong thermal stability of the PNA dimer, potential secondary structures and self-/hetero-dimer formation must be excluded. It can be easily screened with the aid of commercial software or online tools for DNA oligo analysis.

**Statistics**. Statistic significances were calculated by Microsoft Excel 2016 and all the data were shown as mean ± s.d. The two-tailed Student's t test was used to compare differences between two groups with a P value < 0.05 as a threshold for significance. Four technical replicates were performed to improve the statistics.

## Data availability
The data that support the findings of this study are available within the article and its Supplementary Information Files or from the corresponding author upon reasonable request.

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

## Acknowledgements

This work was jointly supported by the National Natural Science Foundation of China (31600595), International Cooperation Project for Science and Research Plan of Shenzhen (GJHZ20160229195805334), Frontier Science Key Programs of Chinese Academy of Sciences (QYZDB-SSWSLH034), Shenzhen Science and Technology Research Funding (JCYJ20160429190215470), Leverhulme Trust (RPG-2015-345 to L.Y.), as well as Hong Kong Research Grants Council (RGC) General Research Funds (GRF) Nos. CityU 11301215 and 11205617.

## Author contributions

W.Z. and X.-F.Y. invented the technique and supervised the project. W.Z. and L.H. performed experiments, analyzed the data, and wrote the paper. Z.Z. contributed to biotinylation of Cas9. L.Y. and P.K.C. contributed to the conception of the project and data review. All authors wrote and edited the manuscript.

## Additional information

**Competing interests:** The authors declare no competing interests.

