## [Peer Review File · Nature Communications]

Reviewers' Comments:

Reviewer #1:

Remarks to the Author:

The powerful CRISPR technique allows the precise and easy genetic manipulation. Recent studies using the CRISPR-associated protein for nucleic acid detection have shown great potential. This work proposes a CRISPR triggered strand displacement amplification method, which makes full use of the advantage of CRISPR for DNA recognizing and binding. The experimental design is based on the conformational rearrangements of Cas9/gRNA recognition and DNA nicking, followed by SDA and PNA invasion-mediated endpoint measurement. The methods present superior sensitivity and single-base specificity in complex background. This is a well-written paper containing interesting results which merit the publication. A number of points need clarifying and certain statements require further justification.

1. At the end of Page 3, the authors mentioned that "all the CRISPR-Dx approaches reported so far require an initial amplification step such as PCR, NASBA and RPA to specifically amplify target nucleic acids and CRISPR effectors are only used in endpoint analyses". In this work, the recognition of the DNA is at the beginning of the experiment and then followed by isothermal amplification of nucleic acid. Could you discuss why the latter is better than the former? Following papers should be cited (J. Am. Chem. Soc., DOI: 10.1021/jacs.8b05309 ; Chem. Sci., 2016, 7, 4951-4957)
2. In Figure 2b and 2d, the sgRNA and IPT1-DNS-Cy3 has "slow migrating complex" and "fast migrating complex", which is confusing. Can you explain this phenomenon?
3. The authors said that "KF polymerase cannot replace the Cas9-sgRNA complex from the binding site during strand elongation and the linear SDA reactions are terminated at the border of opposite Cas9 binding site". Can you provide evidence?
4. In Figure 3. The detection range of hTF1 is 6 orders of magnitude, while the detection of the T1 and T2 region in the human genome covers only 4 orders of magnitude. Can you provide additional data about detecting the T1 and T2 region in the human genome?
5. Supplementary Figure 3 shows significant correlation between the target concentration and detected fluorescence intensity. The fluorescence intensity does not seem to correlate well with the concentration in Figure 3c, 4b and 5b. In addition, in Figure 3c, when the concentration was diluted 10,000-fold from 2.5×10^4 aM to 2.5×10^0 aM, the fluorescence intensity was reduced by about forty percent, while in Figure 3d, when the concentration was diluted 100-fold from 6.7×10^2 aM to 6.7×10^0 aM, the fluorescence intensity was reduced by about sixty percent. Can you explain why?
6. In Supplementary Figure 8, why are there two bands at the position of the amplicon?
7. Supplementary Figure 9 shows "traditional PCR approaches fail to produce any observable amplicons. "Have you considered the possibility that PCR failures are not caused by limitations in detection capabilities, but other reasons such as primer design?"
8. On page 7, "complimentary" should be "complementary".

Reviewer #2:

Remarks to the Author:

This paper by Zhou et al. describes a new method that attempts to detect specified genomic regions from attomolar level of input DNA. The method looks interesting, but a few issues should be addressed.

Major:

- 1, The method requires design of at least 4 sequences within a genomic region of 100 ~ 250 bp, including sgUPS, sgDNS, Biotin-labeled PNA probe and Cy5-labeled probe. To detect a new genomic site, it is hard to design them properly by hand. Could the authors provide a web-tool or other solution to support this method? Also, what proportion of human genome is suitable for

designing using this method?

2, Padlock or MIP(Molecular Inversion Probe) looks more convenient to detect ultra-low amount DNA in isothermal manner. Could the authors give comparisons between this method with Padlock/MIP?

Minor:

3, The last sentence in section of "Ultrasensitive DNA detection by CRISDA" says "PCR approaches at the same concentration fails to produce any observable amplicons (Supplementary Fig. 9)". The authors compared PCR+PAGE-gel with CRISDA, which using SDA + fluorescence. I think it is unfair. The convincing way is either PCR vs SDA in PAGE-gel, or PCR+fluorescence vs CRISDA.

Response to comments from Reviewer #1

We sincerely thank the reviewer for his/her thorough review of our manuscript and for his/her thoughtful comments and valuable suggestions, which are highly helpful in improving this manuscript.

The reviewer's comments are repeated in *italics* and our responses are inserted after each comment.

General Comments

1) The powerful CRISPR technique allows the precise and easy genetic manipulation. Recent studies using the CRISPR-associated protein for nucleic acid detection have shown great potential. This work proposes a CRISPR triggered strand displacement amplification method, which makes full use of the advantage of CRISPR for DNA recognizing and binding. The experimental design is based on the conformational rearrangements of Cas9/gRNA recognition and DNA nicking, followed by SDA and PNA invasion-mediated endpoint measurement. The methods present superior sensitivity and single-base specificity in complex background. This is a well-written paper containing interesting results which merit the publication. A number of points need clarifying and certain statements require further justification.

We appreciate the positive feedback from the reviewer. Our point-by-point reply is presented below.

2) At the end of Page 3, the authors mentioned that “all the CRISRP-Dx approaches reported so far require an initial amplification step such as PCR, NASBA and RPA to specifically amplify target nucleic acids and CRISPR effectors are only used in endpoint analyses”. In this work, the recognition of the DNA is at the beginning of the experiment and then followed by isothermal amplification of nucleic acid. Could you discuss why the latter is better than the former? Following papers should be cited (J. Am. Chem. Soc., DOI: 10.1021/jacs.8b05309; Chem. Sci., 2016, 7, 4951-4957)

In all CRISRP-Dx approaches reported so far, specific amplification of the target nucleic acids by PCR, NASBA, or RPA is a critical precondition to determine the overall performance. Thus, with the exception of searching for new CRISPR effectors with “collateral cleavage activities”, further optimization and extension of CRISRP-Dx approaches are limited. On the other hand, by using CRISPR effectors to trigger subsequent amplification, this approach can be further combined with not only a diverse set of CRISPR effectors with different properties, but also various isothermal amplification techniques to further enhance the robustness, specificity and sensitivity. Thus, CRISDA possesses great potentials to be applied in different scenarios. Indeed, similar strategies have been reported to offer efficient pre-screening of sgRNAs and sensitive *in situ* genomic loci detection when combined with the exponential amplification reaction (EXPAR) and rolling cycle amplification (RCA), respectively.

A sentence discussing this advantage has been added to the main text (Page 4, line 69):

“This unique conformational rearrangement may provide an ideal targeting site for various isothermal amplification techniques with enhanced robustness, specificity and sensitivity due to the intrinsic properties of CRISPR effectors. For example, when combined with the exponential amplification reaction (EXPAR) and rolling cycle amplification (RCA), the CRISPR effector, Cas9, has been successfully applied in the efficient pre-screening of sgRNAs²⁷ and sensitive *in situ* genomic loci detection²⁸, respectively. Thus, this CRISPR effectors-triggered strategy has great potentials to be applied in different situations.”

As suggested by the reviewer, the latest studies *J. Am. Chem. Soc.*, DOI: 10.1021/jacs.8b05309 and *Chem. Sci.*, 2016, 7, 4951-4957 have been cited as references [27] and [28], respectively.

27. Zhang, K.X., Deng, R.J., Li, Y., Zhang, L. & Li, J.H. Cas9 cleavage assay for pre-screening of sgRNAs using nicking triggered isothermal amplification. *Chem Sci* 7, 4951-4957 (2016).

28. Zhang, K. et al. Direct Visualization of Single-Nucleotide Variation in mtDNA Using a CRISPR/Cas9-Mediated Proximity Ligation Assay. *J Am Chem Soc* 140, 11293–11301 (2018).

3) In Figure 2b and 2d, the sgRNA and IPT1-DNS-Cy3 has “slow migrating complex” and “fast migrating complex”, which is confusing. Can you explain this phenomenon?

We agree with the reviewer that the “slow/fast migrating complex” needs to be further clarified in the manuscript.

Since the probe IP_{TI-DNS}-Cy3 is fully complementary to the 5' region of sgRNA (sg_{pTF1-DNS}), they will form a DNA-RNA hybrid. In our EMSA experiments, we notice that there are two complexes of DNA-RNA hybrids with different mobilities. The first complex migrates more slowly than the unbound target DNA (pTF1-Cy5, 269bp), while the other complex migrates much faster than the target DNA (revised **Fig. 2b**). We believe that this phenomenon is caused by the different conformations of sgRNA. The slow migrating complex may represent a hybrid between IP_{TI-DNS}-Cy3 and partially unfolded sgRNA, thus exhibiting a large hydrodynamic radius and slow migration behavior in EMSA. On the other hand, the fast migrating complex may correspond to a DNA-RNA hybrid formed between IP_{TI-DNS}-Cy3 and fully folded sgRNA. Therefore, owing to the small hydrodynamic radius and compact shape, the latter complex migrates fast in EMSA.

In addition, the slow migrating complex disappears as shown in revised **Fig. 2d**. This is because before loading the products to the PAGE gel, we incubate the products at 75 °C for 15 min to inactivate the single-stranded DNA binding protein TP32 (SSB). This heating step leads to complete unfolding of the formerly partially unfolded sgRNA, followed by a re-folding process when the products are transferred to room temperature during the preparation of sample loading. This heating-annealing process facilitates conformational transition and is very common in secondary structures formed by nucleic acids such as G-quadruplexes.

To eliminate confusion, the “slow migrating complex” and “fast migrating complex” are marked by stars in the revised Fig. 2 and explained in the figure caption in the main text (Page 36):

Figure 2. Schematic representation and fluorescent EMSA verification of CRISPR/Cas9-triggered linear SDA. (a) Schematic illustration showing binding of Cy3-labeled IP primer to the exposed region of the nontarget strand in the Cy5-labeled DNA-Cas9 ribonucleoprotein complex. (b) Fluorescent EMSA (6% PAGE) revealing the formation of DNA-Cas9 ribonucleoprotein complex (Species I) and IP-DNA-Cas9 complex (Species II). (c) Schematic illustration showing initiation of linear SDA from the IP primer after adding the SDA mixtures. (d) Fluorescent EMSA (6% PAGE) confirming successful strand elongation from the 3' of IP_{T1-DNS-Cy3} primer to the upstream end of the pTF1-Cy5 fragment. * The slow migrating complex represents the DNA-RNA hybrid formed between IP_{T1-DNS-Cy3} and partially unfolded sg_{pTF1-DNS}, whereas the fast migrating complex represents the hybrid between IP_{T1-DNS-Cy3} and fully folded sg_{pTF1-DNS}.

4) The authors said that “KF polymerase cannot replace the Cas9-sgRNA complex from the binding site during strand elongation and the linear SDA reactions are terminated at the border of opposite Cas9 binding site”. Can you provide evidence?

The first evidence showing such ultra-stability of the DNA-Cas9-sgRNA complex comes from the study investigating interactions between Cas9-sgRNA complexes and DNA targets using DNA curtain assays¹ (cited as reference [34] in the main text). The authors have found that the Cas9-sgRNA complex fails to dissociate from the target site after cleavage unless it is treated with 7 M urea solution.

In addition, we reveal that although a short IP primer is enough to trigger linear SDA reactions (**Fig. 2d** in the main text), a pair of IP primers containing a long 3' overhang complementary to the double-stranded region of the nontarget strand is necessary in order to trigger subsequent exponential SDA reactions (shown in **Supplementary Fig. 5** and discussed on Page 10, line 185 in the main text). This phenomenon clearly shows that the linearly replaced strands are terminated at the border of opposite Cas9 binding site and a long 3' overhang in the IP primer pair is necessary for the initiation of subsequent exponential SDA reactions (illustrated in **Fig. 1**, step 4 in the main text). On the other hand, if elongation of the new strand from IP primer can replace the Cas9-sgRNA complex bound at the opposite site, a shorter IP primer without 3' overhang should be sufficient to trigger subsequent exponential SDA reactions.

Discussion of this phenomenon has been added in the main text (Page 10, line 188):

“First of all, this indicates that KF polymerase cannot replace the Cas9-sgRNA complex from the binding site during strand elongation and the linear SDA reactions are terminated at the border of opposite Cas9 binding site, reflecting the ultra-stability of the DNA-Cas9-sgRNA complex³⁴. If elongation of the new strand from IP primer can replace the Cas9-sgRNA complex bound at the opposite site, an IP primer without the 3' overhang should be sufficient to trigger subsequent exponential SDA reactions.”

It is worth mentioning that although KF polymerase cannot replace the Cas9-sgRNA complex bound at the opposite site, we have found that elevation of the temperature to over 50 °C quickly triggers the dissociation of Cas9-sgRNA complex from the binding site, likely due to unfolding and inactivation of spyCas9 protein at this temperature.

5) In Figure 3. The detection range of hTF1 is 6 orders of magnitude, while the detection of the T1 and T2 region in the human genome covers only 4 orders of magnitude. Can you provide additional data about detecting the T1 and T2 region in the human genome?

We have found that without targeted enrichment of DNA by Cas9, the detection limit of CRISDA towards target DNA is in the attomolar regime. Thus, in the experiments demonstrating the sensitivity of CRISDA, the lowest target concentration is chosen to be 0.1-10 aM. In the case of the hTF1 target, because of its small size (877 bp in length), we can vary the concentration from 2.5 aM to 25 fM to investigate the dynamic range of CRISDA in detecting the target fragments (**Fig. 3c** in the main text). However, in the case of human genomic DNA, because of the large size (6×10^9 bp and ~ 6.1 pg per genome), 1 fM genomic DNA in 20 μ L CRISDA reaction corresponds to 72 ng DNA. Therefore, we have only increased the concentration of human genomic DNA from 6.7 to 670 aM by 4 orders of magnitude including 0 as a negative control (**Fig. 3d**). They correspond to 0.5 ng to 50 ng of human genomic DNA in each 20 μ L CRISDA reaction, because further increases to 500 ng (6.7 fM) or 5 μ g (67 fM) per 20 μ L reaction will be too high for most realistic applications.

Discussion about the different detection ranges towards fragment DNA targets and genomic DNA has been added in the main text (Page 11, line 220):

“Although a detection range between 2.5 aM and 25 fM has been investigated for the target

fragment hTF1 (877 bp), the maximum concentration of genomic DNA in CRISDA amplification is 670 aM, corresponding to 50 ng genomic DNA in every 20 μ L CRISDA reaction. Owing to the large size of human genomic DNA (6×10^9 bp and ~ 6.1 pg per genome), further increase of the human genomic DNA to 500 ng (6.7 fM) or 5 μ g (67 fM) in every 20 μ L reaction will be too high for most realistic applications. Therefore, a detection range of 6 and 4 orders of magnitude (including 0 as a negative control) is used in CRISDA amplification towards fragment DNA targets and genomic DNA, respectively.”

6) Supplementary Figure 3 shows significant correlation between the target concentration and detected fluorescence intensity. The fluorescence intensity does not seem to correlate well with the concentration in Figure 3c, 4b and 5b. In addition, in Figure 3c, when the concentration was diluted 10,000-fold from 2.5×10^4 aM to 2.5×10^0 aM, the fluorescence intensity was reduced by about forty percent, while in Figure 3d, when the concentration was diluted 100-fold from 6.7×10^2 aM to 6.7×10^0 aM, the fluorescence intensity was reduced by about sixty percent. Can you explain why?

We acknowledge that the linear correlation varies among different samples, and it is due to different amplification efficiencies of CRISDA towards different types of targets (target length, sequence complexity and GC-richness). This phenomenon also prevails in PCR-based methods, as it is well-known that it is relatively easy to amplify short and AT-rich templates compared to long and GC-rich ones.

This also explains the second question from the reviewer, since the data in **Fig. 3c** are from the CRISDA reactions using the DNA fragment hTF1 (877 bp) as a template, whereas the data in **Fig. 3d** are from reactions towards human genomic DNA. Although the amplicons (T1 and T2 regions) in **Fig. 3c** and **3d** are the same, human genomic DNA is more complex than the fragment hTF1. When using the human genomic DNA as the template, active Cas9-sgRNA complexes, SSB, KF polymerase and nickase are consumed inevitably at various non-specific sites due to the template complexity. Therefore, the CRISDA reactions give a different dynamic range and amplification efficiency towards human genomic DNA compared to the fragment hTF1.

To explain this phenomenon and to provide strategies for further optimization of CRISDA, the paragraph in the Discussion section has been revised (Page 18, line 363).

“Although spyCas9-mediated CRISDA has great sensitivity and specificity in DNA detection, two issues need to be optimized further in future. First of all, CRISDA exhibits varied dynamic ranges and amplification efficiencies towards different types of targets (target length, sequence complexity and GC-richness). This phenomenon also prevails in PCR-based methods, as it is well-known that long genomic, complex, and GC-rich templates are difficult to amplify⁴⁸⁻⁵⁰. Further experiments are required to optimize the reaction conditions and to choose DNA polymerases with stronger strand displacement activity and higher processivity than KF polymerases. Secondly, nonspecific products from primer dimers are observed especially when the target level is very low. Since the primer dimer is normally generated at a low temperature, further optimization of CRISDA should be carried out using Cas9

ribonucleoproteins working at a higher temperature, for example, AceCas9 and GeoCas9 (from *Geobacillus stearothermophilus*)^{46, 51}. In combination with the thermostable CRISPR effectors, nickases, and DNA polymerases with strong strand displacement activity and high processivity such as the large fragment of *Bst* DNA polymerase, the specificity, sensitivity and efficiency of CRISDA can be improved further to accomplish real-time measurements.”

7) In Supplementary Figure 8, why are there two bands at the position of the amplicon?

The reason for two bands at the position of the amplicon is explained in detail below.

Response Figure 1. Amplification mechanism of CRISDA. (a) Structure of IP primer in CRISDA reactions. (b) Detailed mechanism of the linear and exponential amplification in CRISDA.

As shown in **Response Figure 1a**, a typical IP primer is composed of a 5' overhang serving as a primer after nicking, nickase recognition site, middle hybridization region complementary to the exposed nontarget strand, and 3' overhang complementary to the double-stranded region of the nontarget strand. During amplification (**Response Figure 1b**), the linearly replaced products, Strand-For and Strand-Rev, both contain a truncated nickase recognition site. In subsequent exponential amplification, the Strand-For and Strand-Rev are annealed to IP_{DNS} and IP_{UPS}, respectively, giving final products 1 and 2 with the same length. Afterwards, linearly replaced strands from final products 1 and 2 are annealed to IP_{UPS} and IP_{DNS}, generating products 2 and 1, respectively (the exponential amplification phase). Meanwhile, when the concentrations of products 1 and 2 increase, their linearly replaced products are annealed to each other giving the final product 3. Because product 1 and 2 both have a 5' overhang and a nickase recognition site in one end, they are about 20 bp longer than product 3. Thus, PAGE analysis reveals two bands from the amplicon, where one corresponds to products 1 and 2 with the same length and the other one is product 3, which is 20 bp shorter. This phenomenon has also been observed in conventional SDA reactions as analyzed by PAGE previously (Figure 7

To eliminate the confusion, **Fig. 1** in the main text has been modified in the revised manuscript (Page 35):

Figure 1. Schematic reaction mechanism of CRISDA. **Step 1:** A pair of Cas9 ribonucleoproteins is programmed to recognize each border of the target DNA and to induce a pair of nicks in both nontarget strands. **Step 2:** A pair of IP primers is introduced and hybridized to the exposed nontarget strands. **Step 3:** After adding SDA mixtures containing KF polymerase (3'→5' exo⁻), Nb.BbvCI nikase, and single-stranded DNA binding protein TP32 (SSB), linear SDA is initiated from the binding sites of IP primers, giving linearly replaced single strands, the Strand-For and Strand-Rev. **Step 4:** The products, Strand-For and Strand-Rev, are annealed again to the IP primers, which further induce exponential SDA of the selected target sequence. **Step 5:** The amplicons are quantitatively determined by a PNA invasion-mediated endpoint measurement *via* magnetic pull-down and fluorescence measurements. The well-characterized *S. pyogenes* Cas9 with a mutation of HNH catalytic residue (spyCas9H840A nickase) is used as a model. * Two bands will be observed in PAGE analyses, where one corresponds to the final products 1 and 2 with the same length and the other one is product 3.

The sentences describing this phenomenon have been revised (Page 6, line 114).

“Afterwards, the linearly replaced single strands, Strand-For and Strand-Rev, are annealed to the primers, IP_{DNS} and IP_{UPS}, respectively, which further induce exponential SDA of the selected target sequence within 90 minutes giving products 1 ~ 3 (Step 4). Subsequent PAGE analysis reveals two bands at the position of amplicon, where one corresponds to products 1 and 2 with the same length and the other one is product 3, which is about 20 bp shorter, as has been observed previously³¹.”

8) *Supplementary Figure 9 shows “traditional PCR approaches fail to produce any observable amplicons.” Have you considered the possibility that PCR failures are not caused by limitations in detection capabilities, but other reasons such as primer design?*

We agree with the reviewer that the primer design is very important to the performance of PCR. In our experiments, to fairly compare the performance of PCR and CRISDA, the PCR primer pair (GMO-For/Rev, Supplementary Table 2) is designed within the binding sites of CRISDA IP primer pair (IP_{gTF1-UPS} and IP_{gTF1-DNS}, Supplementary Table 2). Moreover, when designing the GMO-For/Rev primer pair, we carefully consider the length, annealing temperature difference, and possible secondary structures. In addition, as shown in the revised **Supplementary Figs. 9a, b**, this primer pair successfully amplifies 1 ng (25 pM) to 0.1pg (2.5 fM) of target gTF1 and 50 ng (3.66 fM) of GMO genomic DNA diluted without background, confirming that the GMO-For/Rev primer pair is effective for standard PCR. On the other hand, in the presence of interfering DNA and BSA as background, GMO-For/Rev primer pair fails to produce detectable amplicons below 25 fM gTF1 and 3.66 fM GMO genomic DNA (**Supplementary Figs. 9c, d**).

Therefore, we believe that although further optimization of the primer pair or using DNA polymerases with higher fidelity and efficiency than PlatinumTM *Taq* polymerase may improve the performances of PCR reactions, the evidence presented above and in the revised **Supplementary Fig. 9** is sufficient to prove that CRISDA has much better sensitivity than PCR under similar conditions, and the design of PCR primer is not the major reason for the failure of PCR reactions towards attomolar targets.

The following sentences have been added to discuss this issue (Page 12, line 244).

“In comparison, traditional PCR approaches are used to amplify the GMO fragment gTF1 and GMO genomic DNA and the PCR products are subsequently analyzed by PAGE and PNA invasion-mediated endpoint measurements. As shown in Supplementary Figs. 9a, b, the designed GMO-For/Rev primer pair successfully amplifies 1 ng (25 pM) to 0.1pg (2.5 fM) of target gTF1 and 50 ng (3.66 fM) of GMO genomic DNA diluted without background, confirming that the GMO-For/Rev primer pair is effective in standard PCR. However, in the presence of interfering DNA and BSA as the background, PCR fails to produce detectable amplicons below 25 fM gTF1 and 3.66 fM GMO genomic DNA as templates (Supplementary Figs. 9c, d). In addition, only weak fluorescent signals are observed by the PNA invasion-mediated method from the PCR products containing 25 and 2.5 fM gTF1 as templates (Supplementary Figs. 9e, f). The results indicate that the sensitivity of CRISDA is at least three orders of magnitude higher than that of traditional PCR under the same conditions.”

Supplementary Fig. 9 has been modified to (Page 10 in the SUPPLEMENTARY INFORMATION):

Supplementary Figure 9. PAGE analyses and PNA invasion-mediated endpoint measurements towards products amplified by traditional PCR using GMO fragment gTF1 and genomic DNA as templates. PAGE analyses reveal that PCR successfully amplifies (a) 1 ng (25 pM) to 0.1pg (2.5 fM) of target gTF1 and (b) 50 ng (3.66 fM) GMO genomic DNA diluted without background. In the presence of interfering DNA and BSA as background, PCR fails to produce detectable amplicons (c) below 25 fM gTF1 and (d) 3.66 fM GMO genomic DNA as templates. (e) Weak fluorescent signals are observed by the PNA invasion-mediated method from the PCR products containing 25 and 2.5 fM GMO fragment gTF1 as templates. (f) No fluorescence variations are observed from the PCR products using GMO genomic DNA as templates. (Fluorescence signals of CRISDA products are adopted from Fig. 4b and 4c) $n = 4$ technical replicates, two-tailed Student's t test; ** $P < 0.01$, *** $P < 0.001$, **** $P < 0.0001$, bars represent mean \pm s.d.

To describe these experiments, a paragraph has been added to the Methods section (Page 27, line 547).

“PCR-based comparison In comparison, traditional PCR approaches are used to amplify the GMO fragment gTF1 and GMO genomic DNA using primer pair GMO-For/Rev. Ten-fold serial dilutions were performed to prepare target solutions using dilution buffer (10 mM Tris-HCl, 50 mM NaCl, 10 mM MgCl₂, 0.1% Tween 20, pH 8.0) or dilution buffer supplemented with interfering DNA (75 ng μL^{-1} wild type soybean genomic DNA) and BSA (0.5 mg mL⁻¹) as background. The PCR amplifications were carried out using Platinum™ *Taq* DNA polymerase following the manufacturer’s instructions, and the PCR products are subsequently analyzed by 6% native PAGE and PNA invasion-mediated endpoint measurements in the presence of 4 μM SSB.”

9) On page 7, “complimentary” should be “complementary”.

The correction has been made.

References:

- 1 Sternberg, S. H., Redding, S., Jinek, M., Greene, E. C. & Doudna, J. A. DNA interrogation by the CRISPR RNA-guided endonuclease Cas9. *Nature* **507**, 62-67 (2014).
- 2 Joneja, A. & Huang, X. Linear nicking endonuclease-mediated strand-displacement DNA amplification. *Anal. Biochem.* **414**, 58-69 (2011).

Response to comments from Reviewer #2

We sincerely thank the reviewer for his/her thorough review of our manuscript and for his/her thoughtful comments and valuable suggestions, which are highly helpful in improving this manuscript.

The reviewer's comments are repeated in *italics* and our responses are inserted after each comment.

General Comments

1) This paper by Zhou et al. describes a new method that attempts to detect specified genomic regions from attomolar level of input DNA. The method looks interesting, but a few issues should be addressed.

We appreciate the positive feedback from the reviewer. Our point-by-point response is shown below.

Major Comments

2) The method requires design of at least 4 sequences within a genomic region of 100 ~ 250 bp, including sgUPS, sgDNS, Biotin-labeled PNA probe and Cy5-labeled probe. To detect a new genomic site, it is hard to design them properly by hand. Could the authors provide a web-tool or other solution to support this method? Also, what proportion of human genome is suitable for designing using this method?

In our experiments, although three types of sequences, the sgRNAs, IP primer pairs (DNA), and PNA probes, are used in CRISDA reactions, the 3' overhang in IP primer pair is the single critical factor in determining the overall performances of CRISDA. This conclusion is explained below.

In the design of the sgRNA pair directing recognition of Cas9 to target sites and triggering subsequent SDA reactions, we have randomly chosen two 20 bp recognition sites in the forward and reverse strands followed by NGG as the guide and PAM sequences, respectively. By using an online tool (CRISPR Design V1, Zhang's Lab, MIT, 2013), we have reanalyzed these randomly picked sgRNAs applied to detect regions in the human genome. As summarized in **Response Table 1**, these sgRNAs are ranked from "good" (score 85 with only 60 potential off-target sites in the human genome) to "moderate or bad" (score 55 with over 300 potential off-target sites). Interestingly, all these sgRNAs successfully trigger subsequent CRISDA reactions, indicating that sgRNA design is not a critical factor in CRISDA. This phenomenon reflects the sensitivity of Cas9 in recognizing the target DNA, and the advantage of using double Cas9 nickases in triggering exponential SDA reactions to minimize hazardous effects from off-targeting.

Response Table 1. Summary of sgRNAs applied to detect regions in the human genome in this study.

sgRNA Name	Sequence in the guide region (5' ~ 3')	Distance between sg _{UPS} and sg _{DNS}	sgRNA performance analyzed by CRISPR Design (V1)*	
			Score	Number of potential off-target sites in the human genome
sg _{hTF1-DNS1}	CUUGUAGCUACGCCUGUGAU	169 bp	85	60
sg _{hTF1-UPS1}	UUGCAACUGGCCUCAACCUU		77	148
sg _{hTF1-DNS2}	GGCCCAGACUGAGCACGUGA	203 bp	65	275
sg _{hTF1-UPS2}	CCCUUGCUUAAAACUCUCCA		55	312
sg _{hTF2-DNS}	AACUACCCAGUAUUUGUUUC	194 bp	63	243
sg _{hTF2-UPS}	CACAGUUUUAUUCUUCGCUA		74	205

* sequences are analyzed by an online tool, CRISPR Design (V1, Zhang's Lab, MIT, 2013), at: <http://crispr.mit.edu/>.

In the design of the IP primer pair, we have found that the only critical factor is the melting temperature of the 3' overhang complementary to the double-stranded region of the nontarget strand. The 5' sequences containing an overhang (serving as a primer after nicking) and the nickase recognition site are the same in all the IP primers. The middle region hybridized to the exposed non-target strand in the DNA-Cas9-sgRNA complex has limited effects on the performance of the CRISDA reaction. As summarized in **Response Table 2**, the GC-contents of this region in the IP primers used in this study ranged from 68.8% (GC-rich) to 31.2% (AT-rich), with the corresponding melting temperatures between 56.2 and 39.8 °C. This phenomenon confirms our hypothesis that the exposed nontarget strand of DNA targets in the CRISPR ribonucleoprotein complexes may provide an ideal targeting site for various isothermal amplification techniques. On the other hand, the 3' overhang in IP primers is of great importance in triggering exponential SDA and the melting temperature of this region has to be over 50 °C. This issue has been thoroughly discussed in the main text already.

Response Table 2. The GC-content and melting temperature of the middle region in IP primers used in this study.

IP Primer Name	The middle hybridization region complementary to the exposed nontarget strand	
	GC content	T_m
IP _{pTF1-UPS}	62.5%	54 °C
IP _{pTF1-DNS}	68.8%	56.2 °C
IP _{hTF1-UPS1}	56.2 %	52.8 °C
IP _{hTF1-DNS1}	50 %	50 °C
IP _{hTF1-UPS2}	37.5 %	42.6 °C
IP _{hTF1-DNS2}	68.8 %	56.2 °C
IP _{hTF2-UPS}	31.2 %	39.8 °C
IP _{hTF2-DNS}	37.5 %	42.9 °C
IP _{gTF1-UPS}	37.5 %	42.6 °C
IP _{gTF1-DNS}	43.8 %	46.2 °C

In the design of PNA probes targeting the middle region of amplicons, we follow two simple rules. Firstly, melting temperatures of the PNA probes with their target DNA have to be over 37 °C as calculated by the formula established by Giesen *et al*¹. Secondly, because of the strong thermal stability of PNA dimer, the potential secondary structures and self-/hetero-dimer formation have to be excluded. This can be easily screened with the aid of commercial software or online tools for DNA oligo analysis.

Thus, the design of sgRNAs, IP primer pairs (DNA), and PNA probes in CRISDA reactions is much simpler than it looks and we will prepare a web-tool to facilitate the design of CRISDA reactions in the future.

Regarding the second question, we believe that CRISDA may not be suitable for the detection of certain human genomic regions. The first type is the region with extremely high GC contents. Further optimization may include using polymerases with high processive strand-displacing properties, and elevating the reaction temperature with thermal stable CRISPR effectors. Secondly, repetitive genomic regions are not suitable for CRISDA and most of other amplification methods. Thirdly, single-stranded regions in the human genome are not suitable for direct CRISDA detection, because of the absence of nontarget strand to trigger SDA reactions. However, this can be easily solved by introducing an initial strand elongation reaction to synthesize complementary strands or fill single-stranded gaps. Finally, regions with Nb.BbvCI recognition sites in the amplicon are also not suitable for this CRISDA reaction. To overcome this problem, we just need to choose another nickase with different recognition sequences and correspondingly modify the nickase recognition site in the IP primer pair.

A sentence to highlight the simplicity and versatility of CRISDA technique has been added to the Discussion section in the main text (Page 17, line 347):

“On the other hand, the other two components, sgRNAs and PNA probes, both have limited effects in the overall performance of the CRISDA reactions (details are described in the Methods section), thereby making CRISDA a pragmatic technique in the detection of new DNA targets.”

To help readers to understand the simplicity and versatility of the CRISDA technique, a paragraph describing the general considerations of the design of sgRNAs, IP primer pairs (DNA), and PNA probes has been added to the Methods section (Page 29, line 591):

“General considerations for the design of CRISDA sequences In the design of sgRNA pair directing recognition of Cas9 to target sites and triggering subsequent SDA reactions, 20 bp guide sequences followed by NGG as the PAM can be randomly chosen in the forward and reverse strands (100 to 250 bp from each other). In this study, randomly chosen sgRNAs with high specificity (with only 60 potential off-target sites in the human genome) or low specificity (with more than 300 potential off-target sites) all successfully triggered subsequent CRISDA reactions, indicating that sgRNA design is not a critical factor (Supplementary Table 4). In the design of the IP primer pair, the middle region hybridized to the exposed non-target strand in the DNA-Cas9-sgRNA complex had limited effects on the performances of the CRISDA

reaction. As summarized in Supplementary Table 5, the GC-contents of this region in IP primers ranged from 68.8% (GC-rich) to 31.2% (AT-rich), with corresponding melting temperatures between 56.2 and 39.8 °C. The only critical factor in the IP primer was the melting temperature of the 3' overhang complementary to the double-stranded region of the nontarget strand, which has been discussed in details. In the design of PNA probes targeting the middle region of amplicons, two simple rules should be adopted. Firstly, the melting temperature of the PNA probe with target DNA must be over 37 °C. Secondly, because of the strong thermal stability of the PNA dimer, potential secondary structures and self-/hetero-dimer formation must be excluded. It can be easily screened with the aid of commercial software or online tools for DNA oligo analysis.”

Response Table 1 and 2 have been added as Supplementary Table 4 and 5 in the Supplementary information, respectively (Page 21 and 22).

3) Padlock or MIP (Molecular Inversion Probe) looks more convenient to detect ultra-low amount DNA in isothermal manner. Could the authors give comparisons between this method with Padlock/MIP?

The Molecular Inversion Probe² and its ancestor, the padlock probe³, are both single-stranded DNA molecules containing a linker connecting two regions complementary to the target DNA, with a total length of over 80 bases. In combination with conventional PCR or isothermal rolling cycle amplification (RCA), they exhibit great sensitivity and specificity in the detection of nucleic acid targets and SNP genotyping. However, compared to CRISDA method, padlock/MIP methods have certain intrinsic limitations.

Firstly, the padlock/MIP methods require single-stranded DNA targets (ssDNAs) as templates, but CRISDA works perfectly with double-stranded DNA templates (dsDNAs). Since a large number of diagnostic DNA targets are in the double-stranded form, additional pre-treatments are required for efficient detection of these targets by using the padlock/MIP methods. For example, to transform dsDNAs into ssDNAs accessible to padlock/MIP probes, various approaches have been used to pre-treat target dsDNAs, including digestion by restriction enzymes⁴, denaturation by chemical reagents⁵ or heat², and more recently, PNA openers^{6,7}. In particular, when heat denaturation is used to expose ssDNAs, padlock/MIP methods are not truly isothermal. Besides, this pre-treatment step inevitably increases the complexity of the padlock/MIP methods and limits their applications under certain circumstances. On the other hand, unwinding of target dsDNAs in our CRISDA method is facilitated by Cas9 target recognition and cleavage at primer binding sites thereby eliminating the requirements of such pre-treatment step. Thus, compared to the padlock/MIP methods, CRISDA has great potential in applications such as point-of-care diagnostics and field analyses.

Secondly, padlock/MIP methods are not as convenient as CRISDA in detection of long genomic targets. For example, padlock/MIP probes around 120 bases in length have been found to be only suitable for the detection of genomic targets less than 200 bp and to detect a 500 bp target, the optimized padlock/MIP probes as long as over 300 bases in length are required⁸. On account of the complexity and high costs in the synthesis of long ssDNA probes,

padlock/MIP methods are not suitable in detection of targets longer than 200 bp. On the other hand, although we state that the CRISDA method can be used to amplify regions between 100 ~ 250 bp in length (Fig. 1 in the main text), we have also found that it can specifically amplify an amplicon as long as 506 bp down to 2.5 fM (Response Figure 1). In addition, in the CRISDA reactions targeting amplicons with different lengths, relatively short IP primer pairs (~ 60 bases each) are designed by following the simple rule as stated above (T_m of 3' overhang > 50 °C) without further optimization. Thus, CRISDA is more convenient than padlock/MIP methods in the amplification and detection of long genomic targets.

Response Figure 1. Representative PAGE analysis showing CRISDA is able to amplify a 506 bp region in target DNA fragment down to 2.5 fM.

In addition, we believe that CRISDA and Padlock/MIP are two distinct methods since they recognize different types of targets (dsDNA vs. ssDNAs/RNAs) and have different application scenarios (POCT/field analyses vs. multiplexed SNP genotyping/genomic partitioning). Thus, it is inappropriate to experimentally compare these two intrinsically different methods, otherwise we have to compare CRISDA with all other isothermal amplification methods such as conventional SDA, LAMP, EXPAR and so on. On the other hand, since PCR is regarded the gold standard for the detection of nucleic acid targets, we think experimental comparisons between CRISDA and PCR is sufficient in highlighting the advantages of the CRISDA technique.

To compare CRISDA and padlock/MIP methods, sentences have been added to the main text:

“Although several methods such as the padlock/molecular inversion probes (MIP)-mediated methods, nucleic acid sequence-based amplification (NASBA), strand displacement amplification (SDA), loop-mediated isothermal amplification (LAMP), helicase-dependent amplification (HDA), and recombinase polymerase amplification (RPA) have been proposed¹⁻⁷, they suffer from trade-offs with regard to sensitivity, specificity, simplicity, and cost.” (Page 3, line 38)

“In the CRISDA reactions, unwinding of target duplex DNA at primer binding sites is facilitated by Cas9-targeted recognition and cleavage^{25, 26}, thus eliminating the requirements for expensive thermocycler in PCR-based methods or the initial pre-treatment to expose ssDNAs in most isothermal amplification techniques such as the padlock/MIP methods, conventional SDA, and LAMP^{2, 4, 5}.” (Page 16, line 335)

“In contrast, multiple primer pairs or long probes are required and subjected to further

optimization by other isothermal amplification techniques such as the conventional SDA, LAMP and padlock/MIP-mediated methods^{4, 5, 45}.” (Page 17, line 351)

In addition, three representative studies about padlock/MIP methods have been cited as references [1], [2] and [45] in the revised manuscript.

[1] Nilsson, M. et al. Padlock probes: circularizing oligonucleotides for localized DNA detection. *Science* **265**, 2085-2088 (1994).

[2] Hardenbol, P. et al. Multiplexed genotyping with sequence-tagged molecular inversion probes. *Nat. Biotechnol.* **21**, 673-678 (2003).

[45] Krishnakumar, S. et al. A comprehensive assay for targeted multiplex amplification of human DNA sequences. *Proc. Natl. Acad. Sci. U. S. A.* **105**, 9296-9301 (2008).

Minor Comments

4) *The last sentence in section of "Ultrasensitive DNA detection by CRISDA" says "PCR approaches at the same concentration fails to produce any observable amplicons (Supplementary Fig. 9)". The authors compared PCR+PAGE-gel with CRISDA, which using SDA + fluorescence. I think it is unfair. The convincing way is either PCR vs SDA in PAGE-gel, or PCR+fluorescence vs CRISDA.*

We acknowledge that to compare the performance of PCR and CRISDA, direct head-to-head comparisons with comparable methods need to be performed. Therefore, we compared the fluorescence signals of CRISDA and PCR products determined by the PNA invasion-mediated endpoint measurement under the same conditions (for the same concentration of PNA probes and single-stranded DNA binding protein TP32).

As shown in the revised **Supplementary Fig. 9**, although parallel amplification of GMO fragment gTF1 and GMO genomic DNA *via* the traditional PCR approaches at the same concentration fails to produce any observable amplicons as analyzed by PAGE, weak fluorescent signals (determined by the PNA invasion-mediated endpoint measurement) are observed from PCR products containing 25 and 2.5 fM gTF1 as templates and further decrease in the gTF1 concentrations fails to produce any observable fluorescence variations. In addition, no fluorescence variations are observed from the PCR products using GMO genomic DNA as templates. These results indicate that CRISDA is at least three orders of magnitude more sensitive than the traditional PCR under the same conditions.

We have carefully revised the original description in the main text (Page 12, line 244):

“In comparison, traditional PCR approaches are used to amplify the GMO fragment gTF1 and GMO genomic DNA and the PCR products are subsequently analyzed by PAGE and PNA invasion-mediated endpoint measurements. As shown in Supplementary Figs. 9a, b, the designed GMO-For/Rev primer pair successfully amplifies 1 ng (25 pM) to 0.1pg (2.5 fM) of target gTF1 and 50 ng (3.66 fM) of GMO genomic DNA diluted without background, confirming that the GMO-For/Rev primer pair is effective in standard PCR. However, in the presence of interfering DNA and BSA as the background, PCR fails to produce detectable

amplicons below 25 fM gTF1 and 3.66 fM GMO genomic DNA as templates (Supplementary Figs. 9c, d). In addition, only weak fluorescent signals are observed by the PNA invasion-mediated method from the PCR products containing 25 and 2.5 fM gTF1 as templates (Supplementary Figs. 9e, f). The results indicate that the sensitivity of CRISDA is at least three orders of magnitude higher than that of traditional PCR under the same conditions.”

Supplementary Fig. 9 has been modified to (Page 10 in the SUPPLEMENTARY INFORMATION):

Supplementary Figure 9. PAGE analyses and PNA invasion-mediated endpoint measurements towards products amplified by traditional PCR using GMO fragment gTF1 and genomic DNA as templates. PAGE analyses reveal that PCR successfully amplifies (a) 1 ng (25 pM) to 0.1pg (2.5 fM) of target gTF1 and (b) 50 ng (3.66 fM) GMO genomic DNA diluted without background. In the presence of interfering DNA and BSA as background, PCR fails to produce detectable amplicons (c) below 25 fM gTF1 and (d) 3.66 fM GMO genomic DNA as templates. (e) Weak fluorescent signals are observed by the PNA invasion-mediated method from the PCR products containing 25 and 2.5 fM GMO fragment gTF1 as templates. (f) No fluorescence variations are observed from the PCR products using

GMO genomic DNA as templates. (Fluorescence signals of CRISDA products are adopted from Fig. 4b and 4c) $n = 4$ technical replicates, two-tailed Student's t test; ** $P < 0.01$, *** $P < 0.001$, **** $P < 0.0001$, bars represent mean \pm s.d.

To describe these experiments, a paragraph has been added to the Methods section (Page 29, line 547).

“PCR-based comparison In comparison, traditional PCR approaches are used to amplify the GMO fragment gTF1 and GMO genomic DNA using primer pair GMO-For/Rev. Ten-fold serial dilutions were performed to prepare target solutions using dilution buffer (10 mM Tris-HCl, 50 mM NaCl, 10 mM MgCl₂, 0.1% Tween 20, pH 8.0) or dilution buffer supplemented with interfering DNA (75 ng μL^{-1} wild type soybean genomic DNA) and BSA (0.5 mg mL^{-1}) as background. The PCR amplifications were carried out using Platinum™ *Taq* DNA polymerase following the manufacturer's instructions, and the PCR products are subsequently analyzed by 6% native PAGE and PNA invasion-mediated endpoint measurements in the presence of 4 μM SSB.”

References

- 1 Giesen, U. *et al.* A formula for thermal stability (T_m) prediction of PNA/DNA duplexes. *Nucleic Acids Res.* **26**, 5004-5006 (1998).
- 2 Hardenbol, P. *et al.* Multiplexed genotyping with sequence-tagged molecular inversion probes. *Nat. Biotechnol.* **21**, 673-678 (2003).
- 3 Nilsson, M. *et al.* Padlock probes: circularizing oligonucleotides for localized DNA detection. *Science* **265**, 2085-2088 (1994).
- 4 Szemes, M. *et al.* Diagnostic application of padlock probes-multiplex detection of plant pathogens using universal microarrays. *Nucleic Acids Res.* **33** (2005).
- 5 Antson, D. O., Isaksson, A., Landegren, U. & Nilsson, M. PCR-generated padlock probes detect single nucleotide variation in genomic DNA. *Nucleic Acids Res.* **28**, E58 (2000).
- 6 Gomez, A., Miller, N. S. & Smolina, I. Visual detection of bacterial pathogens via PNA-based padlock probe assembly and isothermal amplification of DNAszymes. *Anal. Chem.* **86**, 11992-11998 (2014).
- 7 Yaroslavsky, A. I. & Smolina, I. V. Fluorescence Imaging of Single-Copy DNA Sequences within the Human Genome Using PNA-Directed Padlock Probe Assembly. *Chem. Biol.* **20**, 445-453 (2013).
- 8 Krishnakumar, S. *et al.* A comprehensive assay for targeted multiplex amplification of human DNA sequences. *Proc. Natl. Acad. Sci. U. S. A.* **105**, 9296-9301 (2008).

Reviewers' Comments:

Reviewer #1:

Remarks to the Author:

It has been revised well.

It could be accepted after the author provide the missing Issue and pages of Ref. 28.

Reviewer #2:

Remarks to the Author:

The authors had clarified the advantages and limitations of their new method CRISDA over other existed related methods, and made the comparison between CRISDA and PCR much fairer now.